# Towards Scalable Exact Machine Unlearning Using Parameter-Efficient Fine-Tuning

**Somnath Basu Roy Chowdhury**[1]**, Krzysztof Choromanski**[2,3]**, Arijit Sehanobish**[4]**,
Avinava Dubey**[5]**, Snigdha Chaturvedi**[1]

[1]UNC Chapel Hill, [2]Google DeepMind, [3]Columbia University, [4]Independent,[5]Google Research.
{somnath, snigdha}@cs.unc.edu, {kchoro, avinavadubey}@google.com

## ABSTRACT

Machine unlearning is the process of efficiently removing the influence of a training data instance from a trained machine learning model without retraining it from scratch. A popular subclass of unlearning approaches is *exact machine unlearning*, which focuses on techniques that explicitly guarantee the removal of the influence of a data instance from a model. Exact unlearning approaches use a machine learning model in which individual components are trained on disjoint subsets of the data. During deletion, exact unlearning approaches only retrain the affected components rather than the entire model. While existing approaches reduce retraining costs, it can still be expensive for an organization to retrain a model component as it requires halting a system in production, which leads to service failure and adversely impacts customers. To address these challenges, we introduce an exact unlearning framework – **S**equence-aware **S**harded **S**liced Training ($S^3T$), which is designed to enhance the deletion capabilities of an exact unlearning system while minimizing the impact on model's performance. At the core of $S^3T$, we utilize a lightweight parameter-efficient fine-tuning approach that enables parameter isolation by sequentially training layers with disjoint data *slices*. This enables efficient unlearning by simply deactivating the layers affected by data deletion. Furthermore, to reduce the retraining cost and improve model performance, we train the model on multiple data sequences, which allows $S^3T$ to handle an increased number of deletion requests. Both theoretically and empirically, we demonstrate that $S^3T$ attains superior deletion capabilities and enhanced performance compared to baselines across a wide range of settings.[1]

## 1 Introduction

In recent years, the growing success of machine learning (ML) has led to its widespread deployment across a range of applications (Achiam et al., 2023; Team et al., 2023; Qayyum et al., 2020; Surden, 2021). Once a machine learning model has been trained, it is often necessary to *unlearn* specific training data instances for various reasons, like complying with user data deletion requests (Mantelero, 2013; European Parliament & Council of the European Union; Shastri et al., 2019; Achille et al., 2024), removing stale or corrupt data (Biggio et al., 2012; Steinhardt et al., 2017), etc. Retraining an ML model entirely from scratch with each deletion request is expensive, especially for modern large-scale models (Brown et al., 2020; Achiam et al., 2023; Team et al., 2023). Machine unlearning (Nguyen et al., 2022; Xu et al., 2023) techniques focus on efficiently unlearning the influence of a data instance from a trained machine learning model.

Machine unlearning techniques are classified into two broad categories: approximate and exact unlearning (Xu et al., 2024). Approximate unlearning techniques (Guo et al., 2020; Liu et al., 2024a) modify the parameters of a trained model to reduce the influence of the deleted data instance. While cost-effective in practice, approximate unlearning cannot guarantee the complete removal of an instance's influence and it may still retain non-zero influence on the model. Moreover, auditing approximate unlearning is challenging due to the stochastic nature of ML optimization (Thudi et al., 2022). An alternative to this approach is exact unlearning (Cao & Yang, 2015; Bourtoule et al., 2021; Golatkar et al., 2023), which can guarantee the removal of a data instance's influence from a trained

---

[1]https://github.com/brcsomnath/S3T

model. Exact unlearning techniques use a modular system, where different components within the system are trained on disjoint data subsets (Section 2). When a deletion request occurs, only the affected component needs to be retrained. However, in real-world settings, halting a production system to even retrain a single component can result in service failure and significant expenses. The alternative is to function without the affected component, which may result in reduced performance, ultimately impacting end users. To address these challenges, we introduce **S**equence-aware **S**harded **S**liced Training ($S^3T$), which can exactly unlearn private fine-tuning data efficiently while minimizing the impact on performance.

The key idea behind our $S^3T$ framework is to perform additional offline training before deploying the initial model to reduce retraining costs. At the core of $S^3T$, we leverage a novel lightweight fine-tuning approach that allows parameter isolation by sequentially training model layers using disjoint data slices (Section 3.2). Due to this parameter isolation, it is possible to efficiently perform exact unlearning by deactivating the layers associated with the deleted instance, rather than discarding the entire checkpoint. We efficiently train multiple models using different sequences of the same data slices depending on a training budget. The training budget decides the number of models we can train. We show that increasing the training budget before deployment can significantly reduce retraining costs and improve the model's performance. We also observe that it is important to train the model using diverse sequences and provide several approaches for selecting diverse sequences (Section 3.3). Furthermore, we formalize the notion of deletion rate for exact unlearning systems and theoretically show that $S^3T$ achieves provably better deletion guarantees than existing approaches (Section 3.4). We conduct extensive empirical evaluations to evaluate the effectiveness of $S^3T$ using Transformers with parameter counts ranging from 86M to 13B on a range of tasks. Finally, in Section 4, we empirically validate the sequence selection algorithm and show that $S^3T$ has superior deletion performance compared to existing methods in real-world settings.

## 2 BACKGROUND

Machine unlearning techniques for deep learning is broadly classified into two categories: *approximate* and *exact* unlearning (Xu et al., 2024). Approximate unlearning techniques focus on reducing the influence of a deleted instance from a model after it has been trained. Exact unlearning techniques provide unlearning guarantees by ensuring model components trained on a deleted instance are not used during inference. In this section, we discuss each of these categories in detail.

**Approximate Machine Unlearning**. These techniques focus on approximating the model parameters as if the deleted data instance was not there in the training set from the beginning (Guo et al., 2020). These techniques typically quantify the influence of an instance (Koh & Liang, 2017) and perform gradient ascent for unlearning (Golatkar et al., 2020a;b; Neel et al., 2021; Sekhari et al., 2021; Gupta et al., 2021; Suriyakumar & Wilson, 2022; Liu et al., 2024a). In contrast to these approaches, (Graves et al., 2021) stores the exact gradients encountered during training and uses them directly for gradient ascent. Another line of work (Tarun et al., 2023a;b; Jia et al., 2023; Chen & Yang, 2023; Eldan & Russinovich, 2023; Patil et al., 2023; Kurmanji et al., 2024; Liu et al., 2024b; Park et al., 2024; Zhang et al., 2024) focuses on unlearning in a batch setting, where they assume access to both a retention set and a forget set of data instances for approximate unlearning. While efficient in practice, auditing approximate unlearning techniques is challenging due to the stochastic nature of the optimization process (Thudi et al., 2022; Wang et al., 2024) and may have weak privacy guarantees in practice (Hayes et al., 2024; Łucki et al., 2024).

**Exact Machine Unlearning**. These techniques focus on developing a modular machine learning system, where individual components are trained using disjoint subsets of the data. Such a system offers the advantage that when a deletion request is received for an input instance, we only need to retrain the affected component rather than the entire model. Exact unlearning systems can guarantee unlearning and privacy as none of its components is affected by the deleted instance. However, these systems require modifying the original training process of the model. The seminal work for such a modular unlearning system is Sharded, Isolated, Sliced, and Aggregated training (SISA) (Bourtoule et al., 2021). SISA uses an ensemble of models each trained on a disjoint shard of the dataset as shown in Figure 1 (left). To further reduce retraining costs, each shard is divided into slices, and the models are incrementally trained on these slices, with their checkpoints stored sequentially (shown in Figure 1 (right)). If a deletion request affects a data instance within the 4th slice of a shard, we must retrieve the checkpoint after the 3rd slice and retrain using the remaining data within that shard. Several

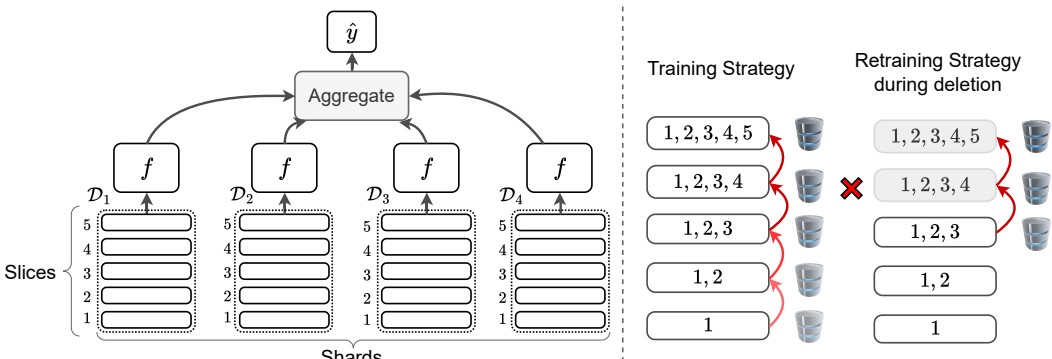

Figure 1: Schematic diagram of the Sharded, Isolated, Sliced, and Aggregated training (SISA) (Bourtoule et al., 2021) framework. An ensemble of models is individually trained on disjoint shards ($\mathcal{D}_i$'s). (*Left*) Each shard is further divided into slices. (*Right*) Each model is sequentially trained on the slices and checkpoints are stored. After deletion, retraining resumes from the best available checkpoint.

approaches focus on improving the components within SISA in application-specific settings like enhancing the dataset partitioning mechanism (Aldaghri et al., 2021; Yan et al., 2022), the retraining efficiency using light-weight adapters (Kumar et al., 2023; Dukler et al., 2023), or extending the modular approach for vision tasks by using compartmentalized diffusion models (Golatkar et al., 2023). However, these approaches are orthogonal to our work since they do not fundamentally modify the functioning of SISA. These modifications can be easily incorporated within our framework.

SISA is a well-known framework that can guarantee exact unlearning and has found widespread applications. However, using SISA within production systems is challenging because retraining even a single component would result in downtime, thereby reducing the service level agreement (SLA). Furthermore, in the worst-case scenario, if deletion requests impact the first slice in all the shards, the entire service goes down, necessitating retraining the model from scratch. In this work, we leverage parameter-efficient fine-tuning to introduce a framework that improves upon the service availability and deletion capabilities of the SISA framework.

## 3    SEQUENCE-AWARE SHARDED SLICED TRAINING (S³T)

In this section, we describe the functioning of our proposed exact unlearning framework, **S**equence-aware **S**harded **S**liced **T**raining (S³T). This framework leverages parameter-efficient fine-tuning (PEFT) to build an exact unlearning framework.

### 3.1    PROBLEM SETTING

We consider the general setting where the user fine-tunes a pre-trained model like BERT (Devlin et al., 2019) or Llama (Touvron et al., 2023) on private data using PEFT techniques. We assume that the deletion requests affect only the private fine-tuning data, not the pre-training data. In S³T, we partition a dataset $\mathcal{D} = \{\mathcal{D}_1, \ldots, \mathcal{D}_m\}$ into $m$ disjoint shards. Each shard is further divided into $L$ slices: $\mathcal{D}_i = \{S_1, \ldots, S_L\}$. S³T trains a separate model per shard and uses their aggregate decision.

Existing unlearning frameworks SISA (Bourtoule et al., 2021) use a similar setup described above. In SISA, within each shard, the model is trained in multiple stages sequentially on the slices (training stages are Slice 1, Slice 1+2, and so on), and their checkpoints are stored. However, a key weakness of SISA is that if deletion requests affect Slice 1 of all shards, then the entire service goes down necessitating retraining from scratch. Another drawback of SISA is that individual models within the ensemble need to be retrained whenever a deletion request is executed. Retraining even on a single slice is expensive for large-scale models in production serving a huge customer base. A naive alternative would be to use the last known usable checkpoint and perform retraining after regular intervals. For example, in Figure 1 (right), if a deletion request arrives for a data instance in Slice 4, the model in production can be replaced with the checkpoint obtained after Slice 3. It is easy to see that the performance of the overall model will degrade with the number of deletion requests.

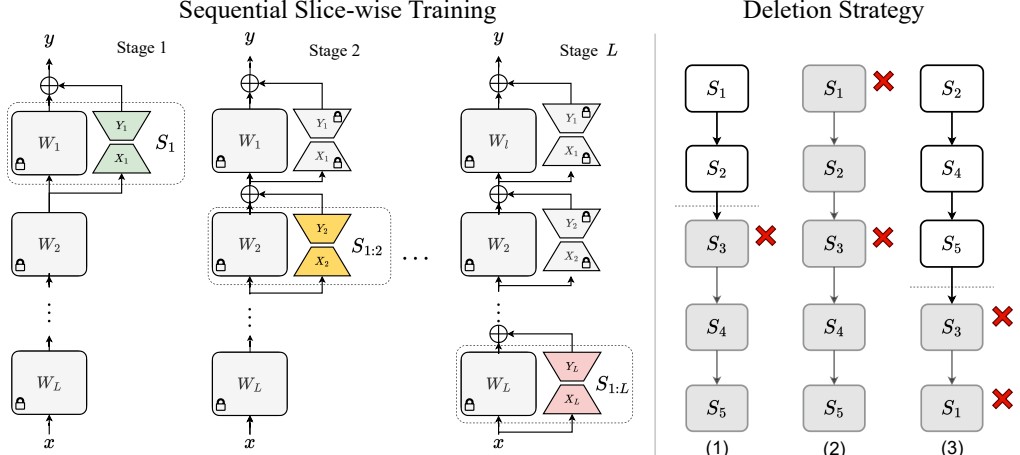

Figure 2: (*Left*) We show the schematic diagram of the slice-wise training strategy in S³T. We incrementally train the model – $i$th layer (from the top) using slices $S_{1:i}$ while keeping the other layers fixed. (*Right*) We show the impact of deletion on models trained on different permutations of slices.

We present an exact unlearning framework, S³T, to address these challenges. The core idea involves training several copies of each model (within the ensemble) using different slice sequences. When deletion requests occur, we utilize the model that minimizes performance degradation. To further reduce the training cost, we leverage PEFT techniques and present a novel sequential slice-wise fine-tuning strategy in Section 3.2. Our training strategy allows us to use the same model by deactivating certain layers without the need to swap checkpoints in case of a deletion. In the following sections, we introduce the fine-tuning strategy and sequence selection process.

## 3.2 SEQUENTIAL SLICE-WISE TRAINING

In this section, we introduce **S**equence-aware **S**harded **S**liced **T**raining (S³T) a lightweight fine-tuning approach using parameter-efficient fine-tuning (PEFT) for efficient exact unlearning. This fine-tuning approach enables parameter isolation by sequentially training PEFT layers using different data slices. Due to this parameter isolation, it is possible to efficiently handle deletion requests by deactivating layers associated with the instance.

We describe S³T using the PEFT technique, LoRA (Hu et al., 2021), but our method is general can be easily extended to other PEFT techniques. LoRA introduces a small number of trainable low-rank $(r \ll d)$ parameters, $(\mathbf{X}, \mathbf{Y})$, while the pre-trained weights $\overline{\mathbf{W}}$ remains fixed as shown below:

$$\mathbf{W} = \overline{\mathbf{W}} + \mathbf{X}^\top \mathbf{Y}, \text{ where } \mathbf{X}, \mathbf{Y} \in \mathbb{R}^{r \times d}, \overline{\mathbf{W}} \in \mathbb{R}^{d \times d}. \tag{1}$$

Our key idea involves training different LoRA layers using different data slices. This approach allows us to selectively deactivate (zero out) the LoRA parameters ($\mathbf{X}$ or $\mathbf{Y}$) associated with a particular layer in the event of data deletion from that slice. In Figure 2 (left), we illustrate the training process in detail, where we follow a sequential top-to-bottom training approach. At stage 1, we train the final model layer (Layer 1 in the figure) using slice 1 while LoRA parameters from all other layers are switched off. In the next stage, we train second last layer (Layer 2) using slices 1 & 2, while keeping the LoRA parameters from the Layer 1 frozen. This process continues for the rest of the layers. Note that the training does not need to proceed in a single layer-wise fashion, we can even train multiple LoRA layers per slice. We discuss more details about the design choice in Appendix C.2.

The sequential slice-wise training process ensures that the LoRA parameter updates at the $i$-th layer are a function of the data instances within slices $\{1, \ldots, i\}$. Therefore, if a deletion request affects the $i$-th slice, the same model can still be used by deactivating the LoRA parameters corresponding to slices $\{i, \ldots, L\}$ (see details in Algorithm 4). For example, if a deletion request affects slice $S_3$ only the subsequent LoRA layers need to be deactivated to ensure exact unlearning, as shown in the first example of Figure 2 (right). This is because during training the parameters of layers 1 & 2 were not affected by instances in $S_3$. During this deletion process, we use the same model checkpoint and switch off LoRA layers, resulting in an $L$-time reduction in storage cost compared to SISA.

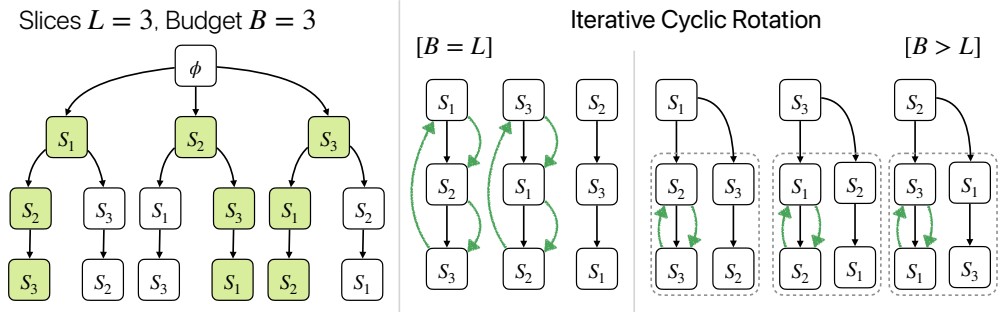

Figure 3: Illustration of the slice sequence selection problem with uniform deletion prior under a budget constraint, $B$. (*Left*) An example of a permutation tree with $L = 3$ and a diverse set of sequences for budget $B = 3$ is shown in green. (*Center*) We show the functioning of the cyclic rotation algorithm, where we generate cyclic permutations of the original sequence. (*Right*) We iteratively extend the algorithm when budget $B > L$ by generating cyclic rotations of the subsequences.

Now we consider the scenario where deletion requests affect multiple slices. This is shown in the 2nd sequence of Figure 2 (right), where slices $S_1$ and $S_3$ are affected. In this case, we observe that a model trained on the default ordering of the sequence $\{S_1, \ldots, S_5\}$ is rendered useless when $S_1$ and $S_3$ are affected. This motivates us to train multiple models using different permutations of the slices. This would enhance the service time and system performance by selecting a model trained with the most effective ordering (e.g., the 3rd sequence in Figure 2 (right) yields the best-performing model). However, training on all $L!$ slice permutations is prohibitively expensive. In the following section 3.3, we present strategies to select a diverse set of permutations under budget constraints. Using these selection approaches, in Section 3.4 we theoretically show that we do not need more than $L$ sequences to achieve the optimal deletion performance.

## 3.3 TRAINING UNDER BUDGET CONSTRAINTS

The training strategy discussed in the previous section requires training the same model on $L!$ slice sequences, which is expensive in practice. In this section, we discuss strategies to select sequences under a budget, $B$ (maximum number of sequences that can trained). First, we show that there exists an optimal subset of $B$ slice sequences and randomly selecting $B$ sequences may not be effective. To illustrate this idea, we use a permutation tree (Bhattacharya, 1994), where all possible permutation sequences are embedded into a tree structure.

In Figure 3 (left), we show an example of a permutation tree with $L = 3$ slices, paths from the root to the leaves correspond to unique permutation sequences $(S_1, S_2, S_3)$, $(S_1, S_3, S_2)$, and so on. From the previous section, we know that the topmost slice is the most sensitive because if a deletion request affects the topmost slice the entire model needs to be retrained (shown in Figure 2 (right)). To address this and reduce the retraining cost, we should ensure we train models on sequences with different top slices. Building on this intuition, in the general setting we should train the model on diverse permutation sequences. Two sequences are considered *diverse if no element appears at the same position*, e.g., $(S_1, S_2, S_3)$ and $(S_2, S_3, S_1)$. An example is illustrated in Figure 3 (left), where a diverse set of 3 sequences is marked in green (where no identical slices occupy the same position). Selecting diverse permutations is challenging as relying solely on random sampling may not always yield the best results. Moreover, in certain scenarios, it is possible to have prior knowledge about the deletion probabilities of data slices; for example, younger users might be more likely to request data deletion than older users. Therefore, we present two strategies for selecting diverse permutation sequences for a budget $B$, depending on whether or not prior deletion probabilities are available.

**Uniform Deletion Prior**. In the setting, where each slice has a uniform (or unknown) deletion prior, we can generate diverse sequences by using cyclic permutations of the original sequence. Given a sequence $(S_1, S_2, S_3)$, the cyclic permutations are (shown in Figure 3 (middle)):

$$(S_1, S_2, S_3) \rightarrow (S_3, S_1, S_2) \rightarrow (S_2, S_3, S_1). \tag{2}$$

The above approach guarantees that no element is repeated in the same position. However, it can only generate up to $L$ different sequences. For budget $B > L$, we extend the cyclic rotation algorithm to generate more sequences. In Figure 3 (right), we generate new sequences by iterating over the

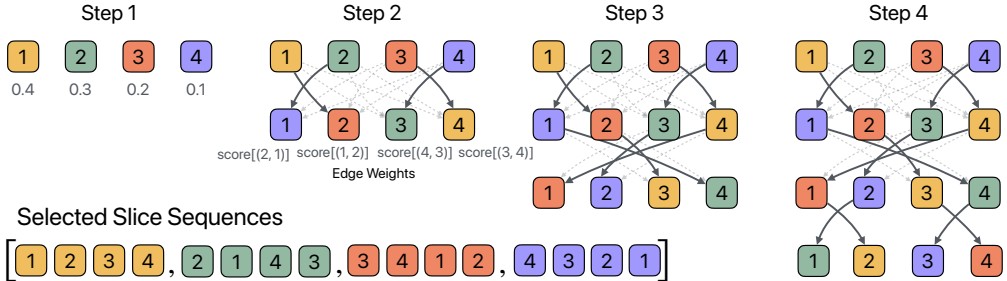

Figure 4: Illustration of the BMS algorithm. BMS selects one element for each permutation at a time. This is done by constructing a bipartite graph with all feasible edges to the next node, where edge weights are the current sequence scores. We compute the maximum weight matching on this graph. The dark gray arrows ($\rightarrow$) indicate the selected edges and dotted arrows ($\dashrightarrow$) the feasible ones.

existing sequences and performing cyclic rotations of the subsequences. For example, for $(S_1, S_2, S_3)$ we perform cyclic rotation of the 2$^{\text{nd}}$ and 3$^{\text{rd}}$ element to obtain the sequence: $(S_1, S_3, S_2)$ (more examples in Figure 10). We provide the general *iterative cyclic rotation* algorithm in Appendix B.1.

**Non-uniform Deletion Prior**. In scenarios, where we have prior knowledge of the deletion probabilities, sequences generated by cyclic rotation may not be ideal. For example, consider the deletion probabilities are: $(S_1: 0.5, S_2: 0.4, S_3: 0.1)$. Then, the first sequence in Eq. 2 is a bad choice because two of the slices most likely to be deleted are placed at the top. It is possible to select better sequences while satisfying the diversity criteria (no repeated slices at the same position). Before describing the selection algorithm, we show how to evaluate the quality of a sequence. We evaluate a sequence with deletion probabilities: $\mathcal{S} = (p_1, \ldots, p_L)$ by computing the expected number of functioning slices after $t$ deletions: $\text{score}[\mathcal{S}, t] = \sum_{i=1}^{L} i \cdot \left(1 - \sum_{j=1}^{i} p_j\right)^t$, where $t$ is a hyperparameter.

We present a bipartite-matching based sequence (BMS) selection algorithm. We will provide an intuitive explanation of the algorithm here and refer to Appendix B.2 for complete details. An illustration of BMS is shown in Figure 4. BMS iteratively selects elements of sequences by constructing a bipartite graph between one level to the next one (where edges are incident on feasible elements for the current sequence). The edges are weighted by the score of the current sequence, $\text{score}[\mathcal{S}, t]$. Selecting the next element then is equivalent to finding a maximum weight perfect matching (Galil, 1986) using the Hungarian algorithm (Kuhn, 1955) shown by the bold lines in Figure 4. This continues till all sequences have $L$ elements. The BMS algorithm can generate up to $L$ sequences. For a budget $B > L$, we use conditional sampling to randomly generate sequences according to their deletion probabilities (see details in Appendix B.2).

**Overall Workflow**. So far, we have discussed the sequential training procedure and the sequence selection approach within a budget. Here, we will bring all of the components together and illustrate the functioning of S$^3$T using a simple example. We consider a setting with $m = 3$ shards, $L = 4$ slices, and budget $B = 4$. Initially, for every shard, the available models are trained on sequences:

$$(1, 2, 3, 4), (4, 1, 2, 3), (3, 4, 1, 2), (2, 3, 4, 1)$$

Next, if a deletion request affects slice 1 for the 3$^{\text{rd}}$ shard. Then, the available models for the 3$^{\text{rd}}$ shard are: $(4), (3, 4), (2, 3, 4)$. Notice how all the PEFT layers at or below slice 1 have been switched off. In this scenario, S$^3$T doesn't perform any retraining but continues to function with the best available model: $(2, 3, 4)$ (as it is trained on maximum slices). If the following deletion requests affect slice 2, then the available models are: $(4), (3, 4)$. S$^3$T continues to function with the best model $(3, 4)$. This continues until no models are available in any shard, at which point S$^3$T is retrained from scratch.

## 3.4 THEORETICAL ANALYSIS

In this section, we theoretically analyze the performance of exact unlearning system. For this, we draw inspiration from approximate unlearning literature (Sekhari et al., 2021) and introduce the definition of *deletion rate* for exact unlearning systems.

**Definition 1** (Deletion Rate). *The deletion rate, $\delta(S)$, of an exact unlearning system $S$, is the expected number of deletion requests until the system needs to be retrained from scratch.*

The deletion rate captures the effectiveness of an exact unlearning system by quantifying the expected number of deletion requests it can handle. Next, we theoretically quantify the deletion rate for $S^3T$ and compare it with that of SISA.

**Lemma 1.** *For uniform deletion prior and dataset size $N \gg r$, where $r$ is the number of deletion requests, the deletion rate of $S^3T$ is $\delta(S^3T) \sim O(mL \log(m \min(B, L)))$ and for SISA it is $\delta(SISA) \sim O(mL \log m)$, where $m$ is the number of shards and $L$ is the number of slices per shard.*

The above result states that the deletion rate doesn't improve by increasing the budget $B$ beyond $L$. This shows that the optimal deletion rate can be achieved using *only $L$ sequences* (instead of $L!$). The complete proof is presented in Appendix A.1.

Next, we analyze the impact of deletion requests on the system's performance. We perform a fine-grained analysis focusing on the performance of an individual shard. In this setting, we consider the real-world scenario where we do not retrain every time a slice is impacted instead work with the best available model. For $S^3T$, this means switching off the necessary PEFT layers, while for SISA, it means reverting to the best available checkpoint. The unlearning system experiences performance degradation with an increasing number of deletion requests, as we are compelled to utilize a model trained on fewer data slices (we show this empirically in Section 4.2). To quantify the performance retention we use a monotonically increasing function $F(k)$, which indicates a model's performance when trained on $k$ slices. The exact formulation of $F(\cdot)$ depends on several factors like the dataset, model size, etc. We analyze the performance retention while processing deletion requests.

**Lemma 2** (Performance Retention)**.** *Given a set of randomly selected $B > 1$ sequences and uniform deletion prior of slices, the difference in the probability that a shard retains a performance, $F_r(\cdot)$ of at least $F(k)$ after $r$ deletion requests, between $S^3T$ and SISA, is shown below*

$$\forall k \in [1 \dots L], \ \mathbb{P}\left[F_r(S^3T) \geq F(k)\right] - \mathbb{P}\left[F_r(SISA) \geq F(k)\right] = \zeta(1 - \zeta^{B'-1}), \quad (3)$$

*where $\zeta = 1 - (1 - k/L)^r$ is a positive fraction and $B' = \min\left\{B, \frac{L!}{(L-k)!}\right\}$.*

The above result demonstrates that compared to SISA, $S^3T$ significantly enhances performance by increasing the probability that the system maintains at least $F(k)$ performance. Eq. 3 shows that increasing the training budget $B$ helps improve the performance of the unlearning system. The proof is presented in Appendix A.2. Another takeaway from the above lemma is that to ensure the system performance of $F(k)$, there is no point in increasing the budget $B$ beyond $\frac{L!}{(L-k)!}$.

## 4 EXPERIMENTS

In this section, we outline the experimental setup and evaluate the unlearning performance of $S^3T$ across various setups. The goal of the experiments is to evaluate the functioning of individual components and the unlearning capabilities of $S^3T$. Specifically, we design experiments to answer the following research questions:

(**RQ1**) Does $S^3T$ training (Section 3.2) impact the model's performance compared to full training?
(**RQ2**) Does $S^3T$ enhance the deletion capabilities of unlearning, and what is its cost tradeoff?
(**RQ3**) Is the sequence permutation selection algorithm (Section 3.3) effective in practice?

### 4.1 SEQUENTIAL SLICE-WISE TRAINING PERFORMANCE

In this section, we evaluate the effectiveness of slice-wise training. The objective of the experimental setup is to demonstrate that $S^3T$ can achieve performance comparable to full training. Note that $S^3T$ is not designed as a new fine-tuning technique that outperforms the baselines. The goal of $S^3T$ is to achieve parameter isolation for data slices without impacting the overall performance. We perform a range of experiments with different Transformer model sizes ranging from 86M up to 13B. We provide the details of the experimental setup and hyperparameters in Appendix C.1.

In Figure 5, we report the performance of fine-tuning on vision, GLUE, and SuperGLUE benchmarks. We use ViT$_{BASE}$ (Dosovitskiy et al., 2020) (for CIFAR10 & CIFAR100 (Krizhevsky et al., 2009)), ViT$_{LARGE}$ (for Tiny ImageNet (Le & Yang, 2015)), and RoBERTa$_{LARGE}$ (Liu et al., 2019) (for

| Model | MMLU | HellaSwag | PIQA | Winogrande | ARC-c | ARC-e | TruthfulQA | OBQA |
|---|---|---|---|---|---|---|---|---|
| Llama2-7B | $41.3_{(0.4)}$ | $76.0_{(0.4)}$ | $79.1_{(1.0)}$ | $69.1_{(1.3)}$ | $46.3_{(1.5)}$ | $74.6_{(0.9)}$ | $39.0_{(1.4)}$ | $44.2_{(2.2)}$ |
| Llama2-7B (FT) | $43.3_{(0.4)}$ | $77.4_{(0.4)}$ | $\mathbf{79.9}_{(0.9)}$ | $69.6_{(1.3)}$ | $50.4_{(1.5)}$ | $\mathbf{74.7}_{(0.9)}$ | $\mathbf{46.3}_{(1.6)}$ | $\mathbf{46.8}_{(2.2)}$ |
| Llama2-7B ($S^3T$) | $\mathbf{43.6}_{(0.4)}$ | $\mathbf{77.7}_{(0.4)}$ | $\underline{79.7}_{(0.9)}$ | $\mathbf{70.5}_{(1.3)}$ | $\mathbf{51.4}_{(1.5)}$ | $\underline{74.1}_{(0.9)}$ | $43.0_{(1.5)}$ | $\underline{46.4}_{(2.2)}$ |
| Llama2-13B | $50.5_{(0.4)}$ | $79.4_{(0.4)}$ | $80.5_{(0.9)}$ | $72.2_{(1.3)}$ | $49.0_{(1.5)}$ | $77.5_{(0.9)}$ | $36.9_{(1.4)}$ | $45.2_{(2.2)}$ |
| Llama2-13B (FT) | $\mathbf{51.8}_{(0.4)}$ | $\mathbf{81.3}_{(0.4)}$ | $\mathbf{82.0}_{(0.9)}$ | $72.2_{(1.3)}$ | $\mathbf{55.0}_{(1.5)}$ | $79.3_{(0.8)}$ | $38.8_{(1.5)}$ | $\mathbf{46.6}_{(2.2)}$ |
| Llama2-13B ($S^3T$) | $\underline{51.6}_{(0.4)}$ | $80.9_{(0.4)}$ | $\underline{81.8}_{(0.9)}$ | $\mathbf{72.8}_{(1.3)}$ | $54.5_{(1.5)}$ | $\mathbf{80.6}_{(0.8)}$ | $\mathbf{39.6}_{(1.5)}$ | $\underline{46.2}_{(2.2)}$ |
| Llama3-8B | $62.2_{(0.4)}$ | $79.1_{(0.4)}$ | $80.6_{(0.9)}$ | $73.6_{(1.2)}$ | $53.2_{(1.5)}$ | $77.6_{(0.9)}$ | $43.9_{(1.4)}$ | $45.0_{(2.2)}$ |
| Llama3-8B (FT) | $59.2_{(0.4)}$ | $\mathbf{80.8}_{(0.4)}$ | $80.9_{(0.9)}$ | $72.2_{(1.3)}$ | $\mathbf{59.0}_{(1.4)}$ | $79.6_{(0.8)}$ | $46.6_{(1.6)}$ | $46.8_{(2.2)}$ |
| Llama3-8B ($S^3T$) | $\mathbf{59.6}_{(0.4)}$ | $\underline{80.2}_{(0.4)}$ | $\mathbf{82.4}_{(0.9)}$ | $\mathbf{73.8}_{(1.2)}$ | $57.0_{(1.5)}$ | $\mathbf{80.5}_{(0.8)}$ | $\mathbf{48.5}_{(1.6)}$ | $\mathbf{47.0}_{(2.2)}$ |

Table 1: Performance comparison of LLMs before and after instruction tuning using Alpaca dataset. We report the performance of Llama2-7B, Llama2-13B, and Llama3-8B models for the following settings: pre-trained model, full training (FT), and slice-wise training ($S^3T$). We observe $S^3T$ achieves comparable performance to FT, even outperforming FT's performance in several datasets. This shows that $S^3T$ allows parameter isolation without affecting the task performance.

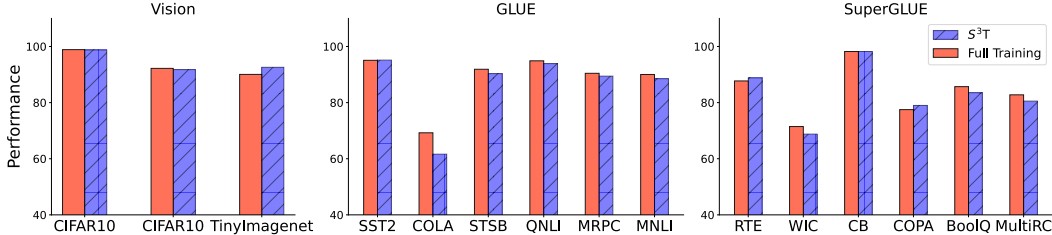

Figure 5: Comparison of the performance between slice-wise training and full training on vision datasets and GLUE & SuperGLUE benchmarks. We report the overall Matthew's correlation for CoLA, Pearson correlation for STS-B, and accuracy for other tasks. We observe that slice-wise training achieves similar performance to full training across all datasets.

GLUE (Wang et al., 2018) & SuperGLUE (Wang et al., 2019)). We observe that $S^3T$ achieves comparable performance to full training (FT) across all settings. In some of the settings, we also observe that $S^3T$ is able to outperform FT (e.g., $S^3T$ obtains 2.5% accuracy gain on TinyImagenet using ViT-large). Next, we conduct experiments to evaluate the effectiveness of $S^3T$ while using large language models (LLMs). Specifically, we perform instruction tuning of Llama2-7B (Touvron et al., 2023), Llama2-13B, and Llama3-8B using Alpaca dataset (Taori et al., 2023). Then, we evaluate each instruction-tuned model on a range of tasks to evaluate the model's general/world knowledge (MMLU (Hendrycks et al., 2020), OpenBookQA (Mihaylov et al., 2018)), truthfulness in QA (TruthfulQA (Lin et al., 2022)), and commonsense reasoning (PIQA (Bisk et al., 2020), HellaSwag (Zellers et al., 2019), Winogrande (Sakaguchi et al., 2021), ARC (Clark et al., 2018)). We use LLM-evaluation suite (Gao et al., 2023) report the zero-shot performance for all datasets. Similar to the previous setup, in Table 1, we observe that $S^3T$ achieves comparable performance to full finetuning across all datasets and even outperforms FT on many datasets across different model sizes. These experiments provide the answer to (**RQ1**) demonstrating that $S^3T$ is an effective way to fine-tune Transformer models, enabling us to achieve parameter isolation for data slices without impacting the model's performance.

## 4.2 DELETION PERFORMANCE

In this section, we evaluate the performance of $S^3T$ as deletion requests are received. In Figure 6 (left), we report the performance of $S^3T$, SISA (Bourtoule et al., 2021), and ProtoSISA (Yan et al., 2022) on CIFAR-10 and CIFAR-100 datasets (all systems use $m = 5$ shards, $L = 4$ slices). In this experiment, we consider a uniform deletion prior over all slices. We also report the performance of full re-training, which retrains the model after each deletion request and serves as an upper performance bound. We report $S^3T$'s performance under various budgets. We observe that $S^3T$ can achieve very close performance to the full budget ($B = 24$) with a significantly smaller budget, $B = 4$. For SISA and ProtoSISA, we observe that the plot ends after approximately 40 deletion requests as these systems do not have functioning models beyond this point. We observe that $S^3T$ can handle more deletion

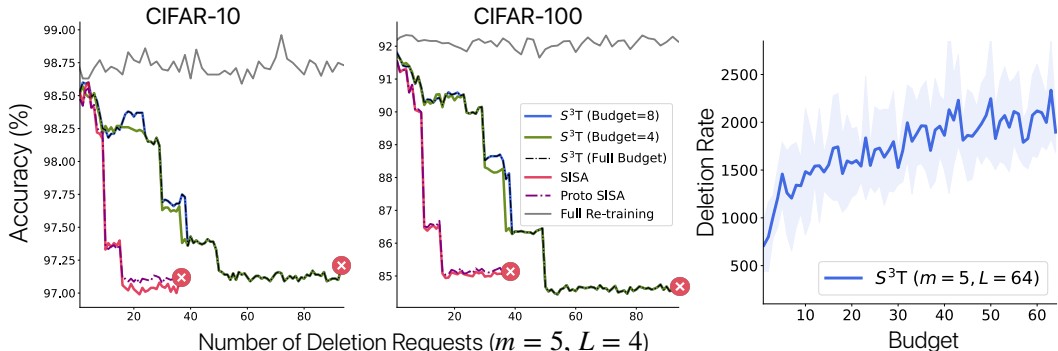

Figure 6: (*Left*) We report the impact on performance of S³T and baselines with an increasing number of deletion requests. We observe that S³T can handle a higher number of deletion requests while maintaining high performance with a relatively low budget (⊗ indicates the failure points). (*Right*) We report the deletion rate of S³T with an increasing budget and observe steady growth.

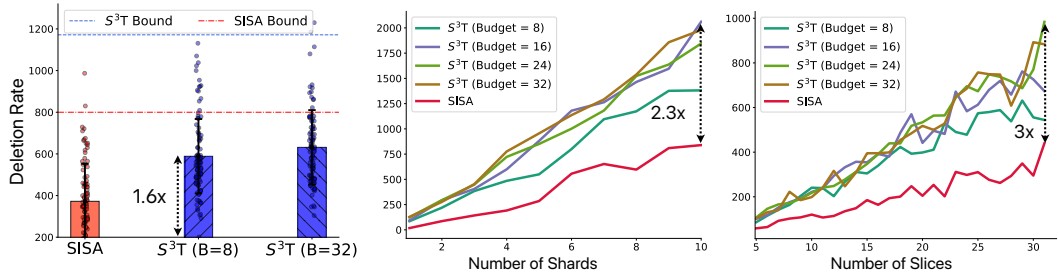

Figure 7: We report the deletion rate of S³T and compare it with baselines. (*Left*) We compare the deletion rates of S³T under varying budgets with SISA and observe significant gains. (*Center*) We report the deletion rates with an increasing number of shards and (*Right*) an increasing number of slices. In both scenarios, we observe that S³T's deletion rate grows significantly faster than SISA.

requests while consistently outperforming the baseline approaches. Note that increasing the budget $B > L$ does not help improve the deletion rate but increases the probability of a better-performing model (as observed in Lemma 1).

Next, we extensively evaluate the impact of increasing the training budget $B$ on the deletion rate (with $m = 5$ shards & $L = 64$ slices). In Figure 6 (right), we observe that there is a steady growth in the deletion rate with an increasing budget. The growth is slightly higher in the initial stages when the budget is low and slows down gradually. This experiment helps answer **(RQ2)** and provides empirical evidence to our theoretical result in Lemma 2, which claims that the performance improves with an increased budget.

## 4.3 ANALYSIS

In this section, we conduct analysis experiments to evaluate different components within S³T. We also report several other analysis experiments in Appendix C.3.

**Deletion Ablations**. We evaluate the deletion capabilities of S³T and compare with the theoretical bounds. In Figure 7 (left), we report the deletion rate of S³T and SISA for the setup (with $m = 5$ shards, $L = 32$ slices). We observe that with a small budget $B = 8$, S³T can achieve 1.6x gains in the number of deletion requests handled. We also plot the theoretical bounds derived in Section 3.4 and show that they hold in practice.

In Figure 7 (center & right), we report the model performance with varying shard and slice counts. As expected, we observe that both increasing the shards and slicing the data more helps in improving the deletion performance. However, the rate of growth in deletion rate is more significant for S³T resulting in up to 2.3x and 3x gains over SISA for the same number of shards and slices respectively.

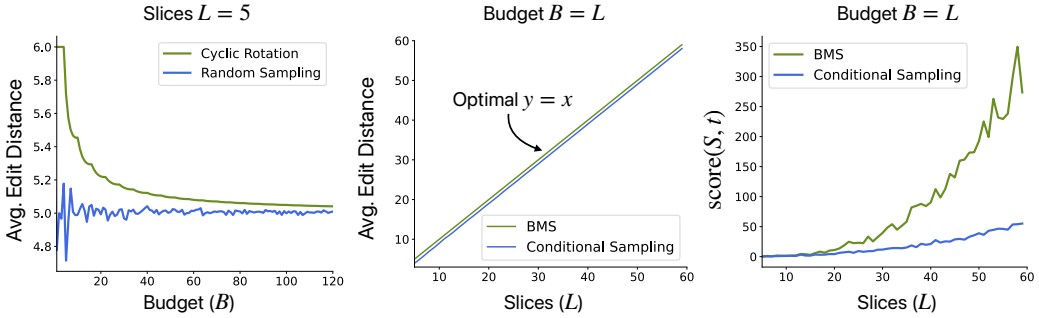

Figure 8: We evaluate the performance of iterative cyclic rotation and bipartite matching-based selection (BMS). (*Left*) We observe that cyclic rotation selection consistently outperforms random sampling for all budgets, $1 \leq B \leq 120$ (with fixed $L = 5$). (*Center*) We evaluate the average edit distance of the sequences generated by BMS and observe that it achieves the optimal edit distance ($L$). (*Right*) We also observe that sequences from BMS achieve higher scores than random sampling.

**Sequence Selection**. We evaluate the quality of the sequences generated by the iterative cyclic rotation and BMS algorithm (Section 3.3). Ideally, we want the selected sequences to be diverse and should have a high edit distance between sequences. Therefore, we report the average edit distance within a selected subset, $O$: $\mathbb{E}_{o \in O}[d_{\text{edit}}(o, o')]$. First, we consider the setting where the deletion prior is uniform and compare cyclic rotation with random sampling. In Figure 8 (left), we report the average edit distance with varying budget ($B$) while the slice count ($L$) is fixed. We observe that cyclic rotation produces significantly better sequences than random sampling. Second, we consider slices associated with a deletion prior. We present the results by averaging over 10 deletion priors sampled from a Dirichlet distribution. In Figure 8 (center & right), we observe that BMS consistently outperforms random sampling both in terms of diversity (avg. edit distance)

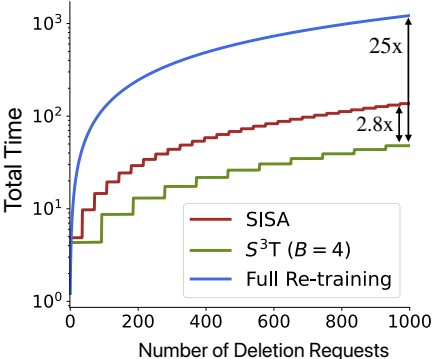

Figure 9: We compare the deletion time as the number of deletion requests increases. $S^3T$ requires the lowest deletion time compared to SISA and full re-training.

and chosen sequence scores ($\text{score}[\mathcal{S}, t]$). These experiments help answer (**RQ3**), showcasing the effectiveness of the proposed sequence selection algorithms in practice.

**Deletion Time**. Unlearning in $S^3T$ is cost-effective; it primarily involves selecting the best checkpoint and swapping it with the current one. The main cost is incurred when a large number of deletion requests necessitate re-training from scratch. In Figure 9, we report the total deletion time (in hours) as the number of deletion requests increases, which includes the checkpoint swapping and re-training time. In this experiment, we compare the deletion time of $S^3T$ with SISA and full re-training on CIFAR10 dataset. Full re-training retrains the model after each deletion request. For SISA and $S^3T$, the step jumps indicate that the system requires retraining. Overall, $S^3T$ achieves the lowest deletion time. $S^3T$ reduces the total deletion time (after 1000 requests) by 2.8x compared to SISA and 25x compared to full retraining. This experiment shows the efficacy of $S^3T$ in reducing the deletion time.

## 5 CONCLUSION

In this paper, we introduced $S^3T$, an effective framework for performing exact unlearning. $S^3T$ uses a modular machine learning model that is trained using disjoint shards of the data. Each shard is used to train a different model using a lightweight fine-tuning approach that enables parameter isolation, which allows us to execute unlearning requests efficiently. The key idea behind $S^3T$ is to train multiple models using different sequences of slices before deploying the system. This helps reduce retraining costs and improve the model's performance while the model is in production. Both theoretically and empirically, we show that $S^3T$ has significantly improved deletion capabilities compared to existing approaches. Future work can focus on developing techniques for finer-grained parameter isolation to further improve $S^3T$'s performance.

## REPRODUCIBILITY STATEMENT

We have submitted the implementation of $S^3T$ in the supplementary materials. We have extensively discussed the details of our experimental setup, datasets, baselines, and hyperparameters in Section 4 and Appendix C. We have also provided the details of our training procedure and sequence selection algorithms in Appendix B. We will make our implementation public once the paper is published.

## ETHICS STATEMENT

In this paper, we introduce an exact unlearning framework, $S^3T$, to enable the efficient unlearning of training data from large-scale models. $S^3T$ has been developed to enable users to remove the impact of private and harmful data from ML models. However, it is crucial to assess whether the deletion requests are initiated by legitimate users. An adversarial user could exploit this feature to degrade the model's performance. In our work, we use publicly available datasets and open-sourced models.

## ACKNOWLEDGMENTS

The authors thank Anneliese Brei, Anvesh Rao, Archiki Prasad, Haoyuan Li, Nicholas Monath, Rajeev Ambati, Rakesh Menon, Shounak Chattopadhyay, and Vinayshekhar Bannihatti Kumar for helpful discussions and feedback on the draft. This work was supported in part by NSF grant DRL-2112635.

## AUTHOR CONTRIBUTIONS

SBRC proposed the unlearning idea, conducted all the experiments, and led the project. KC proposed the diverse permutation selection algorithm using bipartite graph matching. AS contributed to deriving the theoretical results and designing the empirical studies. AD and SC provided ideas and served as project advisors, offering extensive assistance in organizing the manuscript. All authors contributed to the writing of the manuscript.

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

# A    MATHEMATICAL PROOFS

CONTENTS

## A.1    PROOF OF LEMMA 1

In $S^3T$, since the sequences are selected to be diverse (Figure 4), the topmost slice in each sequence is different. Therefore, we have $B' = \min(B, L)$ different slices at the topmost position. This implies that for $S^3T$ to encounter service failure, a total of $mB'$ slices must be affected by deletion requests, where $m$ is the number of shards. Considering deletion requests affect all slices uniformly we need to compute the expected time till all slices are affected. This setup is similar to the coupon collector problem (Blom et al., 1994).

*Proof.* The deletion rate is the expected number of requests to delete all $mB'$ slices ($m$ shards, $B'$ unique slices per shard). Using linearity of expectation, the deletion rate or total time is:

$$\delta(S^3T) = \mathbb{E}[T] = \mathbb{E}[t_1 + \ldots + t_{mB'}] \tag{4}$$
$$= \mathbb{E}[t_1] + \ldots + \mathbb{E}[t_{mB'}], \tag{5}$$

where $\mathbb{E}[t_i] = 1/p_i$, where $p_i$ is the probability that the $i$-th slice is affected after $(i-1)$ slices are deleted. Let $N = |\mathcal{D}|$ the dataset size and $r$ be the number of deletion requests seen so far. The expression of $p_i$ is shown below:

$$p_i = \frac{\{mB' - (i-1)\}s_b}{N - r} \tag{6}$$
$$p_i = \frac{N \min(B/L, 1) - (i-1)s_b}{N - r} \tag{7}$$
$$p_i = \frac{\min(B/L, 1) - (i-1)/(mL)}{1 - r/N} \tag{8}$$
$$\Rightarrow p_i \geq \frac{mB' - (i-1)}{mL}, \tag{9}$$

where $s_b$ is the size of each slice. Replacing this result in Eq. 5, we get:

$$\delta(S^3T) \leq \frac{mL}{mB' - 0} + \frac{mL}{mB' - 1} + \ldots + \frac{mL}{1} \tag{10}$$
$$= mL \left( \frac{1}{1} + \frac{1}{2} + \ldots + \frac{1}{mB'} \right) \tag{11}$$
$$= mL.H_{mB'} \tag{12}$$
$$= mL \log(mB') + \gamma mL + \frac{1}{2} + O\left( \frac{L}{mB'} \right), \tag{13}$$

where $H_{mB'}$ denotes the $mB'$-th Harmonic number and $\gamma$ is the Euler's constant (Lagarias, 2013). The above result proves the first portion of the lemma, $\delta(S^3T) \sim O(mL \log mB')$.

The second part of the lemma is about SISA. For SISA to experience failure, only the first slice of each shard (total of $m$ slices) needs to be affected by deletion. In this case, we can write:

$$p_i = \frac{\{m - (i-1)\}s_b}{N - r} = \frac{N/L - (i-1)s_b}{N - r} = \frac{1 - (i-1)/m}{L(1 - r/N)} \geq \frac{m - (i-1)}{mL} \tag{14}$$

Replacing the above result in Eq. 5, we get:

$$\delta(\text{SISA}) \leq mL\left(\frac{1}{1} + \frac{1}{2} + \ldots + \frac{1}{m}\right) \tag{15}$$

$$= mL.H_m \tag{16}$$

$$= mL\log(m) + \gamma m + \frac{1}{2} + O\left(\frac{1}{m}\right). \tag{17}$$

This proves the second portion of the lemma: $\delta(\text{SISA}) \sim O(mL\log m)$. □

## A.2 PROOF OF LEMMA 2

We begin by proving the retention result for SISA as it is simpler to understand. Each model within SISA is trained on a single sequence of slices.

*Proof.* The probability it maintains performs better than $F(k)$ is equivalent to showing that none of the top-$k$ elements (out of $L$) are affected after $r$ deletion requests:

$$\mathbb{P}\left[F_r(\text{SISA}) \geq F(k)\right] = \left(1 - \frac{k}{L}\right)^r. \tag{18}$$

In S³T, each model is trained on $B$ such sequences. Therefore, we need to compute the probability that at least one sequence is better than $F(k)$, which is:

$$\mathbb{P}\left[F_r(S^3T) \geq F(k)\right] = 1 - \left(1 - \left(1 - \frac{k}{L}\right)^r\right)^B. \tag{19}$$

However, the above result suggests that the probability can be increased indefinitely by increasing $B$. This is inaccurate because to maintain a performance of $F(k)$ there has to be at least one prefix of length $k$ that has not been affected. Since there only $P(L,k) = \frac{L!}{(L-k)!}$ permutations of length $k$ extending the budget beyond $P(L,k)$ doesn't work. Therefore, the correct probability is:

$$\mathbb{P}\left[F_r(S^3T) \geq F(k)\right] = 1 - \left(1 - \left(1 - \frac{k}{L}\right)^r\right)^{B'}, \tag{20}$$

where $B' = \min\{B, P(L,k)\}$.

Taking the difference between Eq. 20 and Eq. 18 and setting $\alpha = (1 - k/L)^r$, we get:

$$\mathbb{P}\left[F_r(S^3T) \geq F(k)\right] - \mathbb{P}\left[F_r(\text{SISA}) \geq F(k)\right] = 1 - (1-\alpha)^{B'} - \alpha$$

$$= (1-\alpha)\{1 - (1-\alpha)^{B'-1}\}$$

$$= \zeta(1 - \zeta^{B'-1}), \tag{21}$$

where $\zeta = 1 - \alpha = 1 - (1 - k/L)^r$. This completes the proof. □

**Discussion**. In the above proof, we assumed that $B$ sequences are randomly selected. In practice, we select diverse sequences using the iterative cyclic rotation algorithm. However, deriving a closed-form theoretical performance bound for the sequences generated using cyclic rotation is non-trivial. Intuitively, we expect the performance to be better as selecting diverse sequences means that the probability of a length-$k$ prefix getting affected is reduced. We leave the theoretical proof of these results to future work.

## A.3 PROOF OF LEMMA 3

This lemma states that BMS selects the most diverse sequences. First, we revisit our definition of sequence diversity. Two sequences are considered diverse if no element appears at the same position, e.g., $(S_1, S_2, S_3, S_4)$ and $(S_2, S_3, S_4, S_1)$. Since we want diverse sequences, BMS performs maximum weight perfect matching, ensuring that no two elements appear in the same position. Therefore, in this proof, we show that perfect matching exists at all levels, $[1, L]$, in Algorithm 2.

*Proof.* Let the bipartite graph, $G$, consist of vertices $V \cup V'$, such that the edges $E \subseteq V \times V'$. For a perfect matching to exist, every subset $W \subset V$ should satisfy:

$$|W| \leq N_G(W), \tag{22}$$

where $N_G(\cdot)$ is the neighbourhood defined using the graph, $G$.

In our setup, the graph $G$ has vertices $V = \{1, \ldots, L\}$ and $V' = \{1', \ldots, L'\}$, which indicate the elements in the permutation sequence. Every vertex $v \in V$ has $(L - i + 1)$ edges to unique vertices in $V'$ at the $i$-th iteration of the algorithm. Therefore, for all iterations $i \in \{2, \ldots, L\}$ (shown in Line 6 in Algorithm 2) the condition in Eq. 22 is satisfied and a perfect matching exists.

Since perfect matching selects a unique element in $V'$, therefore no element is repeated at the same position in the BMS output. Therefore, BMS generates diverse sequences. □

---

**Algorithm 1** Iterative Cyclic Rotation

---

1: **function** CYCLICPERMUTATION(Permutation $P$)
2:     $\mathcal{R} = \{\}$ // set of all rotations
3:     **for** $i \in \{1, \ldots, P.\text{length}\}$ **do**
4:         $\mathcal{R} = \mathcal{R} \cup P$
5:         $P = \text{rotateRight}(P)$
6:     **end for**
7:     **return** $\mathcal{R}$
8: **end function**
9:
10: **function** ITERATIVECYCLICROTATION(Slice count: $L$, Budget: $B$)
11:     $\mathcal{O} = \text{CyclicPermutation}([1, \ldots, L])$ // initialize the set with $L$ cyclic permutations
12:     $n_{\text{iter}} = 0$ // set the iteration count
13:     **while** $|\mathcal{O}| < B$ **do**
14:         $n_{\text{iter}} = n_{\text{iter}} + 1$
15:         **for** $o \in \mathcal{O}$ **do** // iteratively expand each of the existing permutation
16:             $\text{prefix}, \text{suffix} = o[: n_{\text{iter}}], o[n_{\text{iter}} :]$ // set the prefix and suffix
17:             // rotate the suffix with same prefix
18:             $\mathcal{P} = \{\text{prefix} \cup p \text{ for } p \in \text{CyclicPermutation}(\text{suffix})\}$
19:             $\mathcal{O} = \mathcal{O} \cup \mathcal{P}$
20:         **end for**
21:     **end while**
22:     **return** $\mathcal{O}[: B]$ // return $B$ output permutations
23: **end function**

---

# B IMPLEMENTATION DETAILS

In this section, we discuss the details of various algorithms and workflows within S³T.

CONTENTS

## B.1   ITERATIVE CYCLIC ROTATION

In the setting, where the deletion probabilities are uniform we use the cyclic rotation algorithm to select diverse permutations. In Figure 3 (middle), we observe that we can easily generate $L$ diverse sequences using cyclic permutation of the original sequence. We can iteratively expand on this idea to generate more sequences when the budget is $B > L$. An illustration of the different sequences selected using iterative cyclic rotation for different budgets is shown in Figure 10.

We present the pseudocode of the iterative cyclic rotation mechanism in Algorithm 1. First, we set the initial set to cyclic permutations of the original sequence (Line 11). If the budget exceeds the current set of output permutations, we iteratively expand the selected permutations by rotating their suffixes (Line 18). This continues till the output set has at least $B$ sequences. Please note that Algorithm 1 is a simplified version of the actual algorithm. For some corner case budget values, we apply certain heuristics to select the right permutations to expand (Line 18). Empirically, we observe that iterative cyclic permutation significantly outperforms random sampling (Figure 8). We conjecture that iterative cyclic rotation generates the most diverse set of $B$ permutations when all $L$ slices have equal deletion probabilities. We will leave the theoretical proof to future research.

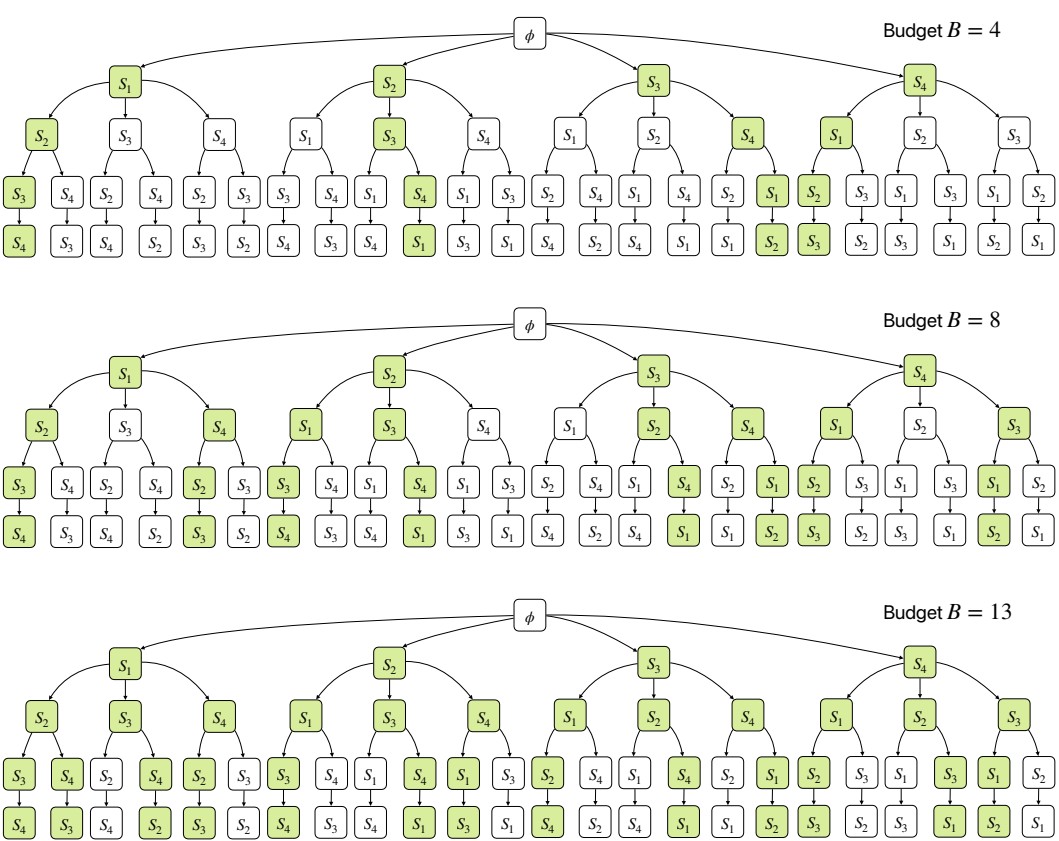

Figure 10: An illustration of the sequences selected by the iterative cyclic rotation algorithm for different budgets. We observe that for budget $B \leq L$ the algorithm selects sequences generated by rotating the entire sequence. For budget $L < B \leq L(L-1)$, the algorithm generates newer sequences by rotating the rotating subsequences starting from the second element. This continues as the budget increases and smaller subsequences are rotated.

## B.2 BIPARTITE-MATCHING BASED SEQUENCE SELECTION

Before describing the bipartite selection algorithm, we discuss the scoring function for sequences with given deletion probabilities. Given a slice sequence with deletion probabilities $\mathcal{S} = (p_1, p_2, p_3, p_4)$, the scoring function computes the expected number of surviving slices after $t$ deletions:

$$\text{score}[\mathcal{S}, t] = 4.(1 - p_1 - p_2 - p_3 - p_4)^t + 3.(1 - p_1 - p_2 - p_3)^t + 2.(1 - p_1 - p_2)^t + (1 - p_1)^t. \quad (23)$$

The above equation is a function of $t$, which needs to be set by the user. In the general case, Eq. 23 can be written as:

$$\text{score}[\mathcal{S}, t] = \sum_{i=1}^{n} i. \left( 1 - \sum_{j=1}^{i} p_j \right)^t. \quad (24)$$

Next, we describe the details of the Bipartite-matching based selection (BMS) algorithm in Algorithm 2. The objective of BMS is to select a set of $B \leq L$ diverse permutation sequences where sequences have a high score (Eq. 24). This algorithm starts by selecting $L$ different starting elements for the sequences in Line 4. Then, BMS iteratively selects the next element within each sequence. This involves constructing a bipartite graph between the last elements of the sequences seen so far and the next set of elements. The edge weights are set to the score of the sequence that is a concatenation of the current sequence, $o$, and the next node, $v'$. The edges incident on feasible next elements (elements not seen in a permutation sequence so far) as shown in Line 12. We perform a maximum weight perfect graph matching on the graph, $G$, using the Hungarian algorithm (Kuhn, 1955). Based

---

**Algorithm 2** BMS Sequence Selection Algorithm

---

1: **function** BMS(Slice count: $L$, Budget: $B$)
2:     $\mathcal{O} = \{\}$
3:     **for** $l \in \{1, \ldots, L\}$ **do** // initializing the sequence set with first element
4:         $\mathcal{O} = \mathcal{O} \cup \{l\}$
5:     **end for**
6:     **for** $i \in \{2, \ldots, L\}$ **do** // iterations to select the $2^{\text{nd}}$ to the $L$-th element
7:         $G = \{\}$ // Initialize Graph
8:         **for** $o \in \mathcal{O}$ **do** // Iterate over sequences
9:             $v = o.\text{pop}()$ // Collect final element of each sequence
10:             **for** $v' \in \{1, \ldots, L\}$ **do** // Iterate and add all feasible edges
11:                 $w = \text{score}[o \cup v']$ // compute score for the sequence $o \cup v'$ (Eq. 24)
12:                 **if** $v' \notin o$ **then** $G.\text{add\_edge}(v, v', w)$ // check for feasibility
13:             **end for**
14:         **end for**
15:         $(V, V') = \text{MaximumWeightMatching}(G)$ // use Hungarian algorithm (Kuhn, 1955)
16:         **for** $o \in \mathcal{O}$ **do**
17:             // select vertex associated with each sequence from perfect matching
18:             $v^* = \{v'|v' \in V' \wedge v = o.\text{pop}()\}$
19:             $o = o \cup v^*$ // add to sequence
20:         **end for**
21:     **end for**
22:     $\mathcal{O}_B = \{o \in \mathcal{O}|o \text{ is among top } B \text{ sequence with highest score}[o]\}$
23:     **return** $\mathcal{O}_B$
24: **end function**

---

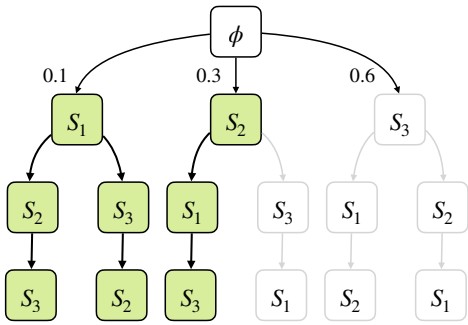

Figure 11: Illustration of conditional sampling of slice sequences. For each sequence, one slice is sampled at a time based on their deletion probabilities. We observe that conditional sampling selects sequences that has slices with relatively lower deletion probabilities towards the top.

on the matching, we select the next element for each permutation sequence (Line 18). BMS continues this process till all sequences $o \in \mathcal{O}$ have $L$ elements. Based on the budget, we output $B$ sequences from $\mathcal{O}$ that have the highest sequence scores. For maximum weight perfect matching, (Tomizawa, 1971) provides an efficient algorithm with time complexity $O(n^3)$, where $n$ is the number of nodes. Using this algorithm, the overall time complexity of BMS is $O(L^4)$, where $L$ is the number of slices. The following result shows that BMS generates the most diverse sequence set.

**Lemma 3.** *For a budget $B \leq L$, BMS returns the most diverse set of $B$ permutations.*

We also validate the above result empirically and find that BMS produces a range of sequences with high scores (Figure 8). However, it remains an open question whether these outputs are optimal (while considering both diversity and scores).

The BMS algorithm can output up to $L$ diverse permutations. Extending it for a budget $B > L$ is non-trivial. Therefore, in this setting, we use a conditional sampling approach to generate diverse sequences. In conditional sampling, for each sequence, we sample slices one at a time based on their deletion probabilities (low deletion probabilities have a higher chance of being sampled). If a

---

**Algorithm 3** S$^3$T Training Procedure

---

1: **Input**: Dataset $\mathcal{D}$, Shard Count: $m$, Slice Count: $L$, Budget: $B$
2: $\mathcal{F} = \{\}$ // initializing model set
3: **for** $d \in \{\mathcal{D}_1, \ldots, \mathcal{D}_m\}$ **do** // partition the dataset into shards and iterate over them
4:     $\mathcal{S} = \{S_1, \ldots, S_L\}$ // partition into slices, $\cup_i \mathcal{S} = d$
5:     $\Pi = \text{SelectTrainingSequences}(L, B)$ // using cyclic or BMS algorithm
6:     **for** $\pi \in \Pi$ **do**
7:         $\mathcal{S}_\pi = \pi(\mathcal{S})$ // order the slices according to the permutation $\pi$
8:         $f = \text{SliceWiseTraining}(\mathcal{S}_\pi)$
9:         $\mathcal{F} = \mathcal{F} \cup f$
10:     **end for**
11: **end for**
12: **return** $\mathcal{F}$

---

**Algorithm 4** S$^3$T Deletion Procedure

---

1: **Input**: Model set $\mathcal{F}$, Deletion instance: $x$
2: $m' = \text{LocateShard}(x)$ // get original shard ID, $m'$
3: $\overline{\mathcal{F}} = \{\}$ // set of modified models
4: **for** $f \in \mathcal{F}_{m'}$ **do** // iterate over models trained on $m'$-th shard
5:     $l' = \text{LocateSlice}(x, f)$ // get original slice ID, $l'$, of $x$ within $f$
6:     **for** $l \in \{l', \ldots, L\}$ **do**
7:         $\text{DeactivateLayer}(f, l)$ // deactivate PEFT layers
8:     **end for**
9:     **if** $l' > 0$ **then** $\overline{\mathcal{F}} = \overline{\mathcal{F}} \cup f$ // ensuring all layers aren't switched off
10: **end for**
11: $\mathcal{F} = \mathcal{F} \setminus \mathcal{F}_{m'}$ // remove older models
12: $\mathcal{F} = \mathcal{F} \cup \overline{\mathcal{F}}$ // introduce updated models
13: **return** $\mathcal{F}$

---

sampled sequence already exists in the selected set, it is discarded. An illustration of the conditional sampling approach is shown in Figure 11. We observe that the selected samples (shown in green) have slices with relatively lower deletion probabilities at the top.

### B.3    S$^3$T TRAINING PROCEDURE

In this section, we provide an outline for the training procedure within the S$^3$T framework in Algorithm 3. S$^3$T proceeds by dividing the entire dataset into $m$ shards. Each shard is further divided into $L$ disjoint slices. Based on the budget $B$, we obtain the permutation sequences to perform slice-wise training. We train each model $f$ on a unique sequence of slice sequence, $\pi(\mathcal{S})$. We return the complete set of models, $\mathcal{F}$. During inference, the user selects the best-performing model within each shard and deploys an ensemble of those models to production.

### B.4    DELETION PROCEDURE

In this section, we describe the procedure when a deletion request arrives for an instance $x$ in Algorithm 4. We first locate the shard ID, $m'$, where the instance $x$ belonged and iterate over all models trained on that shard. For each of these models, we locate the slice ID, $l'$, of $x$ and deactivate all layers $\{l', \ldots, L\}$ (as shown in Line 7). After that, if the model $f$ is still active, we add it to the set of modified models, $\overline{\mathcal{F}}$. S$^3$T uses the updated set to perform inference till a new deletion request arrives. Note that the deletion process can occur entirely offline without impacting the production system, provided that the deleted instance does not influence any of the ensemble models.

Table 2: We report the hyperparameters used in our experiments for fine-tuning S³T across different benchmarks and Transformer model variants.

| Dataset | Model | # Slices (L) | # Layers/ Slice | LoRA Rank (r) | LoRA (α) | Learning Rate | Epochs |
|---|---|---|---|---|---|---|---|
| **GLUE Benchmark** | | | | | | | |
| SST-2 | RoBERTa_LARGE | 7 | 3 | 16 | 32 | $10^{-5}$ | 10 |
| COLA | RoBERTa_LARGE | 7 | 2 | 8 | 16 | $4.10^{-4}$ | 30 |
| STS-B | RoBERTa_LARGE | 7 | 3 | 16 | 32 | $10^{-4}$ | 30 |
| QNLI | RoBERTa_LARGE | 7 | 3 | 8 | 16 | $5.10^{-5}$ | 30 |
| QQP | RoBERTa_LARGE | 7 | 3 | 16 | 32 | $5.10^{-5}$ | 30 |
| MRPC | RoBERTa_LARGE | 7 | 3 | 16 | 32 | $5.10^{-5}$ | 30 |
| MNLI | RoBERTa_LARGE | 7 | 3 | 16 | 32 | $10^{-4}$ | 30 |
| **SuperGLUE Benchmark** | | | | | | | |
| RTE | RoBERTa_LARGE | 8 | 4 | 16 | 32 | $10^{-5}$ | 30 |
| WIC | RoBERTa_LARGE | 12 | 2 | 16 | 32 | $10^{-4}$ | 30 |
| CB | RoBERTa_LARGE | 12 | 2 | 16 | 32 | $10^{-4}$ | 30 |
| COPA | RoBERTa_LARGE | 12 | 2 | 32 | 64 | $10^{-5}$ | 30 |
| BoolQ | RoBERTa_LARGE | 7 | 3 | 16 | 32 | $10^{-4}$ | 30 |
| MultiRC | RoBERTa_LARGE | 7 | 3 | 16 | 32 | $10^{-4}$ | 30 |
| **Vision Benchmark** | | | | | | | |
| CIFAR-10 | ViT_BASE | 6 | 2 | 16 | 32 | $2.10^{-3}$ | 15 |
| CIFAR-100 | ViT_BASE | 6 | 2 | 16 | 32 | $2.10^{-3}$ | 15 |
| TinyImagenet | ViT_LARGE | 6 | 4 | 16 | 32 | $2.10^{-3}$ | 15 |
| **Instruction Tuning** | | | | | | | |
| Alpaca | Llama2-7B | 4 | 8 | 32 | 64 | $2.10^{-5}$ | 3 |
| Alpaca | Llama2-13B | 4 | 10 | 32 | 64 | $2.10^{-5}$ | 3 |
| Alpaca | Llama3-8B | 4 | 8 | 32 | 64 | $2.10^{-5}$ | 3 |

# C  Experiments

In this section, we describe our experimental setup and present additional analysis experiments to evaluate the functioning of S³T.

## C.1  Experimental Setup

We perform all experiments using PyTorch (Paszke et al., 2019) and Huggingface (Wolf et al., 2019) framework. Our experiments were run on NVIDIA A6000 GPUs. In Table 2, we report the common set of hyperparameters for S³T fine-tuning experiments. All hyperparameters were set using a grid search with the Weights & Biases framework. We use an AdamW optimizer with the corresponding learning rates for each dataset (reported in Table 2). During fine-tuning of the models, we perform full-precision training for all settings except instruction tuning where we use 8-bit training.

## C.2  Design Choices

We discuss the rationale behind the design choices for the proposed slice-wise training approach. First, we used top-to-bottom fine-tuning because we found the top layers were easier to train at the start and a bottom-up approach didn't converge. Note that our training process does not need to proceed in a layer-wise fashion, we can even train multiple LoRA layers per slice. Second, we train layers in a cumulative manner where the $i$-th layer is trained using slices $\{1, \dots, i\}$. We found that this way of training helps in better convergence compared to the setup where we train every layer using a different slice. Third, we analyze the cost of slice-wise training and find it to be comparable to full training. Considering that training cost is $c = O(nl)$, where $n$ is the dataset size and $l$ is the number of trainable layers (empirical evidence in Appendix C.3). For slice-wise training, we observe that $c = \sum_{i=1}^{k} \left(\frac{in}{k}\right)\left(\frac{l}{k}\right) = O\left(\frac{nl}{2}\right)$, which is of the same order as full training.

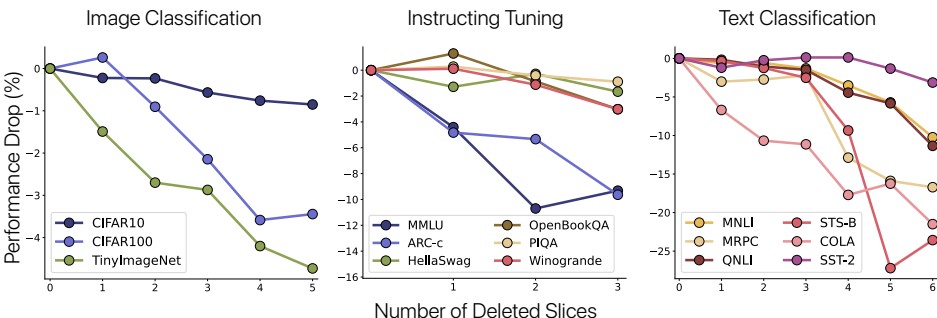

Figure 12: Relative performance drop with an increasing number of deleted slices. We observe a relatively small performance drop for image classification and instruction tuning tasks while noticing a considerable drop for a few of the text classification datasets.

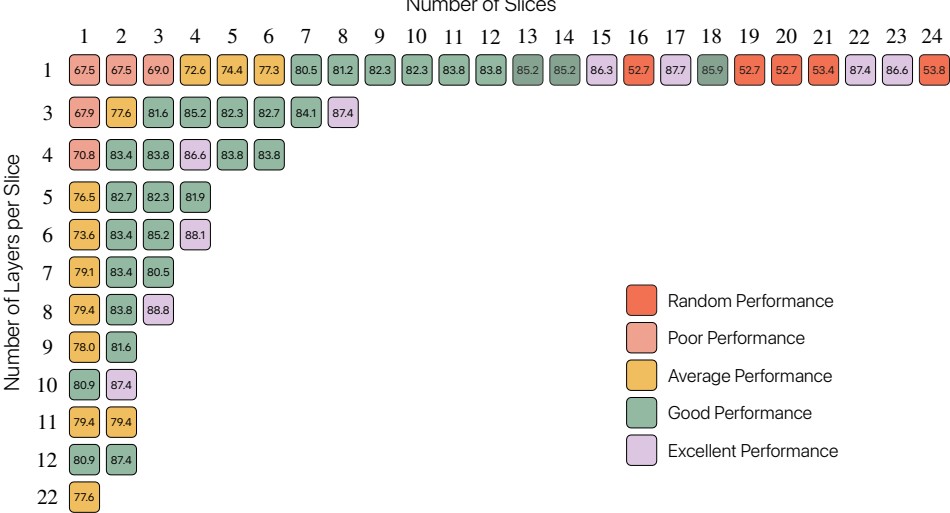

Figure 13: Performance impact of S³T when allocated a different number of layers per slice: We observe that simply increasing the number of slices can lead to a significant performance drop. There is an optimal tradeoff between the total number of slices and the number of layers per slice.

## C.3 ADDITIONAL EXPERIMENTS

In this section, we report analysis experiments to evaluate the S³T's deletion capabilities and training.

**Performance Degradation with Slice Deletion**. In Figure 12, we report the relative performance drop with an increasing number of slices affected by deletion requests. For vision datasets in Figure 12 (left), we only observe a small drop in performance (<5%). For instruction tuning in Figure 12 (middle), we observe a negligible performance drop for most datasets except MMLU & ARC, which are more challenging open-ended QA benchmarks. For text classification in Figure 12 (right), we observe an increased drop in performance for a few of the datasets. We hypothesize that this occurs because the RoBERTa_LARGE (used in text classification tasks) is a relatively weaker pre-trained model compared to Llama and ViT. We also observe that the performance can vary based on the task and overall it is dependent on both the task and the model being fine-tuned. For unlearning, a lower performance drop is better as it suggests that we can continue using the same model without retraining for a longer time.

**Layer Allocation per Slice**. This experiment aims to answer the question: how many slices can we pack into a model while still achieving good performance? We conduct an experiment where we allocate different numbers of PEFT layers per slice and also vary the number of slices. We report the results on the RTE dataset using RoBERTa_LARGE in Figure 13. We observe that the performance is

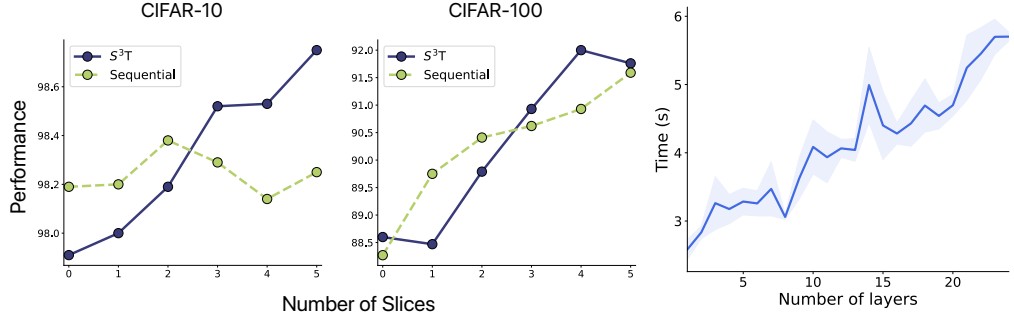

Figure 14: (*Left*) Comparison of sequential training with S³T using a PEFT model. We observe that S³T's performance gradually improves as it is trained on a larger number of slices, ultimately achieving similar performance to sequential training while being more efficient.

| Budget | Model (GB) | PEFT (GB) | Del. Rate |
|--------|-----------|-----------|-----------|
| $B = 1$ | 1.2 | 0.21 | 66.4 |
| $B = 2$ | 1.2 | 0.42 | 86.9 |
| $B = 3$ | 1.2 | 0.63 | 100.8 |
| $B = 4$ | 1.2 | 0.84 | 107.8 |
| $B = 5$ | 1.2 | 1.05 | 110.1 |
| $B = 6$ | 1.2 | 1.26 | 124.1 |

Table 3: Storage cost of S³T under different training budgets. We show how the deletion rate increases with an increased storage cost. The overall storage cost of PEFT layers is minimal and is equivalent to storing an additional model.

poor when the number of layers per slice is low. For example, when the number of layers per slice = 1, the performance improves as we increase the number of slices but again drops when the slice count is too high. This shows that it may not be feasible to train the model using many slices without a drop in performance. Overall, the performance variation depends on the underlying task and model, so the developer should select these hyperparameters based on their performance requirements.

**Sequence-based Training**. In this experiment, we compare S³T's performance with sequential training. Sequential training involves training the entire PEFT model on a sequence of slices. This was used in (Kumar et al., 2023) to improve the retraining efficiency of SISA. In Figure 14, we report the performance of S³T and sequential training using ViT$_{\text{BASE}}$ on CIFAR-10 and CIFAR-100 datasets. We observe that the performance of S³T gradually increases as it is trained on more slices. Overall, we observe that S³T achieves quite similar or outperforms sequential training. It is important to note that S³T trains a significantly smaller number of parameters ($L$ times reduction compared to sequential training) but still achieves competitive performance.

**Storage Costs**. In this experiment, we evaluate how the storage cost of S³T grows with an increasing budget and its effect on the overall deletion rate. In Table 3, we report the storage costs and deletion rate of training ViT$_{\text{LARGE}}$ model with $m = 5$ shards and $L = 6$ slices per shard. We observe that the storage cost of PEFT layers (with LoRA rank=16) is considerably less compared to the full model size. As the budget and storage cost increases there is an improvement in the deletion rate. However, the rate of improvement of the deletion rate slows down with an increased budget, indicating there is a lesser return on increasing the budget.

**Training Time**. In this experiment, we evaluate the training time with a varying number of PEFT layers. In Figure 14 (right), we report the average training time over a constant number of steps using RoBERTa$_{\text{LARGE}}$ model. We observe that training time linearly increases as an increased number of LoRA layers are trained. This shows the effectiveness of our proposed S³T framework, which only trains a small number of PEFT layers at each training stage.

**BMS vs. Cyclic rotation with Deletion prior**. In this setting, we compare the BMS selection algorithm with a variant of cyclic rotation when the prior deletion probabilities are available. We

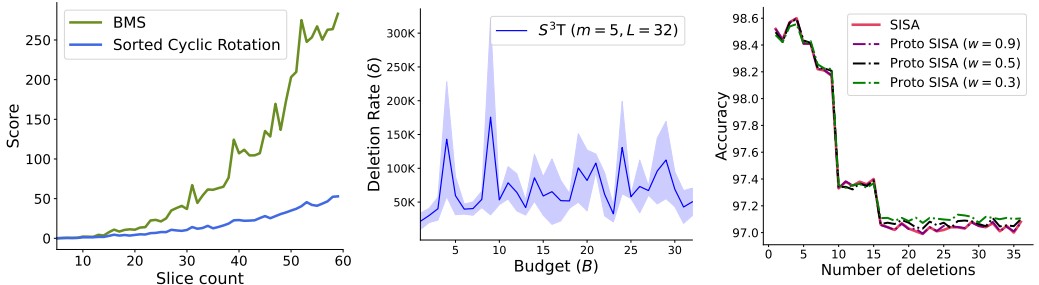

Figure 15: (*Left*) We compare the scores (Eq. 24) of the generated sequences by BMS and sorted cyclic rotation. We observe BMS consistently outperforms cyclic rotation. (*Center*) We show the variation in deletion rate with increasing budget $B$ under non-uniform deletion prior over the slices. (*Right*) We report the performance of SISA and protoSISA with varying weights.

perform cyclic rotation by first sorting the slices based on their deletion probabilities. For example, we sort the sequence with deletion probabilities: ($S_1$: 0.5, $S_2$: 0.4, $S_3$: 0.1) as: ($S_3, S_2, S_1$). Then, for a budget $B = 3$, the stored sequences are: ($S_3, S_2, S_1$), ($S_1, S_3, S_2$), ($S_2, S_1, S_3$). In this variant, the slices most likely to be deleted are not at the top of any sequences. We follow the experimental setup in Section 4.3 and sample deletion priors using a Dirichlet distribution (over 10 runs). In Figure 15 (left), we report the total sequence scores (Eq. 24) obtained by BMS and sorted cyclic rotation for a budget, $B = L$. We observe that BMS consistently outperforms sorted cyclic rotation for all budgets. Please note that the average edit distance achieved by both methods is the same as cyclic rotations are guaranteed to produce the maximum diversity for budgets: $B \leq L$.

**Deletion with Non-uniform Prior**. In this setting, we evaluate the impact of increasing the budget $B$ when the deletion prior is non-uniform. We use $m = 5$ shards and $L = 32$ slices. For each shard, we sample a deletion prior over the slices distributions from a Dirichlet distribution. Using these priors, we select sequences to train the model using the BMS algorithm. Since the BMS algorithm always generates $L$ sequences, we greedily select the top $B$ sequences based on the sequence score defined in Eq. 24. In Figure 15 (center), we report the deletion rate of S³T under varying budget sizes, $B$, when the deletion requests are sampled from the prior. We observe that when we have access to the deletion prior, S³T can handle significantly more deletion requests (compared to the uniform prior setting in Figure 6 (right)). Overall, the deletion rate shows a gradual increase as the budget rises; however, it remains notably high even with a low budget.

**Baseline Ablations**. In this setting, we investigate the performance of ProtoSISA under different settings. ProtoSISA uses the same setup as SISA along with a prototypical classifier. During inference, we use a weighted combination of the logits of SISA and the prototypical classifier. In this experiment, we vary the weight assigned to the SISA component and report the results in Figure 15 (right). We observe that ProtoSISA achieves slightly better performance with lower weights when the number of deletion requests is high.

**Ensemble of Slice Sequences**. In this experiment, we evaluate the impact of using an ensemble of models trained on different slice sequences. These slice sequences include data within the same shard. We use a ViT-base model on the CIFAR100 dataset ($m = 5, L = 4$). In Figure 16, we observe that increasing the number of models initially enhances performance; however, this improvement plateaus as the model count increases.

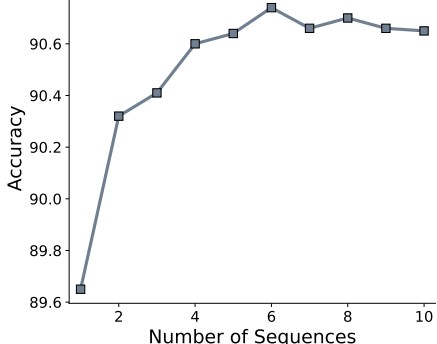

Figure 16: We report the model performance using an ensemble approach where individual models are trained on different slice sequences. We observe an initial improvement in performance, which eventually plateaus.

## D   LIMITATIONS & FUTURE DIRECTIONS

In this paper, we present a novel exact unlearning framework, $S^3T$. $S^3T$ improves the deletion rate of existing unlearning systems at the cost of additional offline training. The training process can be time-consuming if we are fine-tuning larger models and have a higher performance requirement (thereby high budget $B$). However, this is an inherent tradeoff between offline training and losing revenue due to re-training costs. The developer should adjust their budget according to the tradeoff in their specific application.

The primary goal of an exact unlearning system is to maintain a finely modular structure, enabling efficient retraining of individual components upon data deletion. Therefore, it is important to have sample efficient learning methods so that components achieve high performance while using less data. To further improve $S^3T$, this would involve research to develop more expressive parameter-efficient fine-tuning approaches that can perform well when trained with small amounts of data.

## E   BROADER IMPACT

We present a scalable approach to perform exact unlearning in production systems. We hope that this framework will be adopted by organizations and enable a cost-effective way to ensure the privacy requests of various users. In general, unlearning systems are susceptible to attacks where the adversary may design deletion requests in a way to modify the model behavior according to their needs. Therefore, it is important to ensure that the deletion requests executed on the unlearning system are not malicious. However, this is a limitation of unlearning systems in general and not specific to our proposed framework, $S^3T$. Future works can focus on the identification of malicious unlearning requests.

