# OpenReview forum: "Towards Scalable Exact Machine Unlearning Using Parameter-Efficient Fine-Tuning"
_ICLR.cc/2025/Conference — ICLR 2025 Poster_

### Official Review · Reviewer_Masp · 2024-11-02

**Soundness:** 4
**Presentation:** 3
**Contribution:** 3
**Rating:** 8
**Confidence:** 3

**Summary:**

This paper presents a new framework for exact machine unlearning, called S$^3$T, inspired from SISA. S$^3$T leverages PEFT on disjoint shards of data. Different from SISA, S$^3$T sequentially trains PEFT layers following a slice sequence. By using multiple slice sequences, S$^3$T provides granular isolation and superior deletion capabilities, enhancing the deletion rate while maintaining good performance. The authors propose two algorithms to choose slice sequences, for uniform and non-uniform deletion priors. Extensive experiments and ablations on several vision and language benchmarks demonstrate the effectiveness of S$^3$T in comparison to SOTA baselines.

**Strengths:**

- This is a well written paper.
- The paper focuses on a very relevant setting involving PEFT for unlearning.
- Rethinking slicing from SISA into a layerwise training while leveraging PEFT is a clever design. It was interesting to see that such a design does not introduce significant trade-offs.
- The experimental evaluation is extensive, spanning several datasets and model sizes. The authors cover all the important questions relevant to S$^3$T in their evaluations, including its training performance, deletion performance and ablation studies. The diversity and scale of experiments strengthens the contributions of the paper.
- The appendix discusses several important aspects of the work including storage costs, training time, design choices, etc. In particular, it was interesting to note why the top-down approach was chosen over a bottom-up approach.

**Weaknesses:**

- Depending upon the model and the dataset, S$^3$T can result in lower training performance compared to full training.
- No code is provided, however the authors promise to release it with the final version.

The paper can be further improved by adding a limitations section that could summarize all potential trade-offs and discussing future directions to improve the same. It would also be nice to see some performance numbers in the abstract itself, such as S$^3$T improves the deletion rate by $1.6\times$ over SISA.

**Questions:**

1) Are results in Table 1 with B = 1? When B > 1, will using an ensemble of models across different slice sequences enhance performance?
2) On similar lines as the previous question, will using an ensemble of available models improve performance over just using the model trained on the maximum number of slices pertaining to a deletion request?
3) How does the net storage cost of S$^3$T compare to SISA when B > L and B < L?
4) The performance of ProtoSISA seems to be almost overlapping with SISA? Why is this so?

---

> ### Author Response · Authors · 2024-11-18
>
> We would like to thank the reviewer for their detailed comments and suggestions.
>
> > *No code is provided, however, the authors promise to release it with the final version*.
>
> We have submitted the code along with this submission. Please let us know if you have any issues accessing it on Openreview.
>
> > *Include limitations and discuss future directions.*
>
> Thank you for the suggestion. We have included a limitation and future direction section in Appendix D of the paper.
>
> >  *Are results in Table 1 with B = 1? When B > 1, will using an ensemble of models across different slice sequences enhance performance?*
>
> The results in Table 1 have $m=1$ shards and $B=1$ because we are comparing the performance of our sequential training with full training on the entire data.
>
> Thank you for the suggestion. We perform a small-scale experiment to evaluate the impact of using an ensemble of models trained on different slice sequences. We report the results in Appendix C.3 (Ensemble of Slice Sequences). We observe that increasing the number of models improves performance initially, but this improvement plateaus as the model count increases. We do not use an ensemble of models within a shard to maintain a low inference cost and have a lightweight system.
>
> > *will using an ensemble of available models improve performance over just using the model trained on the maximum number of slices pertaining to a deletion request?*
>
> We evaluate this scenario by using an ensemble of models trained on slice sequences of different lengths. We report the results in the table below.
>
>
> Combination of Models w/ Slice Sequences|Performance|
> --|--|
> Ensemble Model w/ slices: 3|89.12|
> Ensemble Model w/ slices: 3+2|89.19|
> Ensemble Model w/ slices: 3+2+1|89.07|
>
>
> Specifically, we consider a ViT-base model trained on different slice sequences. We observe that a model trained with 3 slices achieves a similar performance to an ensemble of models trained with 3 and 2 slices. Including a model trained with a single slice hurts the overall performance. Therefore, using an ensemble of models instead of the best performing model may not be a good idea in the unlearning framework.
>
>
> > *How does the net storage cost of S3T compare to SISA when B > L and B < L*?
>
> In $S^3$T, we need to store $B$ models per shard. Therefore, the total storage cost is $mB$, which grows linearly with the budget $B$.
>
> > *The performance of ProtoSISA seems to be almost overlapping with SISA? Why is this so?*
>
> ProtoSISA uses a prototypical classifier along with SISA. Our experiments using the ViT-base transformer show that both SISA and the prototypical classifier achieved similar performance. Moreover, we use a weighted combination of the decisions of each of the predictors (SISA and prototypical) to obtain the final prediction.
>
> In our current experiments, we assigned a weight of 0.9 to SISA and 0.1 to the prototypical classifier, which leads to similar results. We have added new experiments (Appendix C.2, Baseline Ablations) with SISA weights of 0.5 and 0.3. In these settings, we observe that ProtoSISA’s performance varies from SISA. Specifically, ProtoSISA can boost SISA’s performance when the number of deletion requests is high.

---

> > ### Comment · Reviewer_Masp · 2024-11-20
> >
> > Thank you for the new experiments. The results with the ensemble were interesting to note. The authors also provide code, strengthening further the quality of the submission.
> >
> > I recommend including the best version of ProtoSISA in the main section (Figure 6) for the final version.
> >
> > The authors’ responses have well addressed my questions, and I will maintain my score.

---

> > > ### Author Response · Authors · 2024-11-20
> > >
> > > Thank you for taking the time to review our responses. We have now updated Figure 6 to include the best ProtoSISA results.

---

### Official Review · Reviewer_U5Bx · 2024-11-03

**Soundness:** 3
**Presentation:** 3
**Contribution:** 2
**Rating:** 8
**Confidence:** 3

**Summary:**

The paper tackles machine unlearning,  the process of efficiently removing the influence of a specific training data instance from a machine learning model without retraining it entirely. There are two main types of unlearning: approximate and exact. This paper focuses on exact machine unlearning, a category that guarantees the complete removal of a data instance's influence from the model. Exact unlearning methods rely on training individual components of the model on distinct, disjoint subsets of the data.

The authors introduce a new framework for exact unlearning called Sequence-aware Sharded Sliced Training ($S^3T$). $S^3T$ is designed to improve the ability to delete data instances from a model while minimizing any negative impact on the model's overall performance. Both theoretical and empirical results show that $S^3T$ outperforms existing methods in terms of deletion capability and performance across a variety of scenarios.

The key innovation of the $S^3T$ framework is the use of additional offline training before the model is deployed, which significantly reduces retraining costs. The authors compare their approach with Sharded, Isolated, Sliced, and Aggregated (SISA) training. A major drawback of SISA is that if deletion requests target Slice 1 across all shards, the entire system becomes compromised, necessitating complete retraining. The S3T framework addresses this issue by introducing more robust deletion strategies.

Additionally, the authors use LoRa for slicewise training and present methods for selecting diverse permutations under budget constraints.

**Strengths:**

- Theoretically shown that S3T achieves provably better deletion guarantees than existing approaches.

- Empirical results show that S3T is an effective method for fine-tuning Transformer models, enabling parameter isolation for data slices without compromising the model's performance.

**Weaknesses:**

- The paper exhibits minor weaknesses due to lack of clarity and ambiguity; for detailed points, please refer to the questions provided.

**Questions:**

- What does "Total time" refer to in Figure 9, and in which units is it measured?
- Could you elaborate on the setting where budget B < L?
- Could you explicitly provide answers to RQ2 and RQ3 in the text, similar to how RQ1 is addressed?
- In Figure 1, it’s unclear that the shards are disjoint—can you clarify this?

---

> ### Author Response · Authors · 2024-11-18
>
> We would like to thank the reviewer for their detailed comments and suggestions.
>
> > *What does "Total time" refer to in Figure 9, and in which units is it measured?*
>
> The total time in Figure 9 refers to the overall time spent processing $n$ unlearning requests. It is reported in hours. It includes both the retraining and checkpoint swapping time. We have mentioned this in the paper now.
>
> > *Could you elaborate on the setting where budget B < L?*
>
> Thank you for the question. We believe this question is about how the sequence selection is performed when $B < L$.
>
> Uniform prior. In this case, we generate $L$ sequences using the iterative cyclic rotation algorithm. For $B<L$, we can choose any $B$ out of the $L$ generated sequences as they are equally diverse and have maximum edit distance among each other.
>
> Known prior. In this case, we generate $L$ sequences using the BMS algorithm. For $B<L$, we greedily select the top $B$ sequences with the maximum $score[\mathcal{S}, t]$.
>
> > *Could you explicitly provide answers to RQ2 and RQ3 in the text, similar to how RQ1 is addressed?*
>
>
> Please find the responses below:
>
> (RQ2): We find the S3T significantly improves the deletion rate of existing unlearning systems (as shown in the results of Figure 6). We also observe that the overall unlearning time of S3T is much better than the baselines (Section 4.3, Deletion Time). S3T requires more offline training and needs to train $B$ models per shard.
>
> (RQ3): The sequence selection algorithms presented in the paper are effective. They perform significantly better than naively sampling sequences both in terms of sequence quality (when deletion prior is available) and diversity. The detailed results are shown in Figure 8 (Section 4.3).
>
> We have incorporated these conclusions in Section 4.2 and Section 4.3 respectively.
>
> > *In Figure 1, it’s unclear that the shards are disjoint—can you clarify this*?
>
> Thank you for pointing this out. The shards shown in the figure are disjoint. We have updated Figure 1 and its label to showcase the change.

---

> > ### Author Response · Authors · 2024-11-20
> >
> > Thanks for taking the time to review our responses. Please let us know if you have any questions or would like any further clarifications. If your questions have been answered, we kindly ask that you consider adjusting your score. Thank you!

---

> > > ### Author Response · Authors · 2024-11-24
> > >
> > > Thank you for reviewing our responses again. Since the discussion period ends soon, we wanted to double-check whether your questions have been addressed. If you need any further clarification, please let us know.

---

> > > > ### Author Response · Authors · 2024-11-25
> > > >
> > > > Since the rebuttal period is drawing to a close, we wanted to check in again and see whether our rebuttal has addressed your comments. Please let us know if we can answer any other questions you have.

---

> > > > > ### Comment · Reviewer_U5Bx · 2024-11-26
> > > > > **Clarification Acknowledgement**
> > > > >
> > > > > Thank you for your clarifications. I have addressed my score.

---

> > > > > > ### Author Response · Authors · 2024-11-26
> > > > > >
> > > > > > Thank you again for taking the time to review our responses and updating the score.

---

### Official Review · Reviewer_tCTF · 2024-11-04

**Soundness:** 3
**Presentation:** 3
**Contribution:** 3
**Rating:** 6
**Confidence:** 3

**Summary:**

This paper proposes S3T, a new solution for machine unlearning. Different from Shared, Isolated, Sliced, and Aggregated training (SISA) techniques, S3T allows isolation between parameters by sequentially training model layers using disjoint data slices. S3T trains the pre-trained model multiple times with different training orders to avoid re-training from scratch. The experimental results show that S3T can effectively reduce the re-training parameters after receiving the deletion request, compared with SISA methods.

**Strengths:**

++ This paper points out a strong limitation of current SISA-based machine unlearning methods: the deletion request of the bottom slices will lead to retraining all the above slices. S3T’s solution effectively reduces the re-training time cost.

++ The LoRA-based training solution makes S3T practical in application as it is compatible with many Transformer-Based models, including small and large models.

++ The theoretical analysis of S3T’s deletion rate and performance retention metrics is rigorous.

++ The writings and figures of the paper are easy to follow.

**Weaknesses:**

-- In Section 2, the paper claims that “approximate unlearning techniques… may have weak privacy guarantees in practice.” However, it does not explain whether S3T has a stronger privacy guarantee mechanism.

-- Although S3T outperforms other SISA-based methods, it significantly increases the training burden of the model providers (Even after applying the permutation-tree-based method, S3T still requires training the same model multiple times). I think the paper should further explain whether it is worth increasing the training cost several times or even dozens of times to reduce the cost of re-training, especially in the context of training LLMs.

-- The sequence-order selecting algorithm in Section 3.3 depends on the deletion rate of each slice. In other words, its performance is related to the distribution of each slice’s deletion request’s arrival probability. Therefore, In Section 4.2’s evaluation, the uniform deletion prior over all slices may not be enough to fully evaluate the algorithm’s performance.  The paper could conduct another experiment in which the slice’s arrival probability is more unpredictable.

**Questions:**

Please refer to the comments in Weaknesses.

---

> ### Author Response · Authors · 2024-11-18
>
> We would like to thank the reviewer for their detailed comments and suggestions.
>
> > *Explain how S3T has stronger privacy guarantees*
>
> $S^3$T is an exact unlearning technique. Exact unlearning techniques function by eliminating or retraining all model components that use the unlearnt instances. Since these techniques do not use any component trained on the unlearnt instance, they provide stronger privacy guarantees.
>
> $S^3$T performs unlearning by switching off PEFT layers trained using the unlearnt request. Therefore, the model at any point does not have any component that was trained using the unlearnt instance. This is how $S^3$T provides strong privacy guarantees.
>
> We have added a line to explain this in Section 2, Lines 98-99.
>
>
> > *Explain the utility of increased training cost to reduce re-training in the context of LLMs*
>
> Minimizing retraining costs for production systems is crucial, as it requires taking the system offline every time a deletion request is spawned, directly impacting consumers. Moreover, if the deletion requests affect the same slice, it can lead to redundant re-training.
>
> Even though we are training multiple models, this process can be easily parallelized and it is quite fast in practice since we only train PEFT parameters in Transformer models. In contrast to this, the re-training process is sequential and it is dependent on the deletion request.
>
> > *The paper could conduct another experiment in which the slice’s arrival probability is more unpredictable.*
>
> We have added a new experiment to evaluate the deletion rate of $S^3$T when the deletion requests are sampled from a non-uniform prior. Please see the details of the setup in Appendix C.3 (Deletion with non-uniform prior). We observe that $S^3$T can handle significantly more deletion requests compared to the uniform prior, as the prior is known. Overall, we observe that the deletion rate shows a gradual increase as the budget rises; however, it remains notably high even with a low budget.

---

> > ### Author Response · Authors · 2024-11-20
> >
> > Thanks for taking the time to review our responses. Please let us know if you have any questions or would like any further clarifications. If your questions have been answered, we kindly ask that you consider adjusting your score. Thank you!

---

> > > ### Author Response · Authors · 2024-11-24
> > >
> > > Thank you for reviewing our responses again. Since the discussion period ends soon, we wanted to double-check whether your questions have been addressed. If you need any further clarification, please let us know.

---

> > > > ### Author Response · Authors · 2024-11-25
> > > >
> > > > Since the rebuttal period is drawing to a close, we wanted to check in again and see whether our rebuttal has addressed your comments. Please let us know if we can answer any other questions you have.

---

> > > > > ### Comment · Reviewer_tCTF · 2024-11-26
> > > > >
> > > > > Thanks for the further clarification, however, which has not fully addressed my concerns. I will keep my current rating.

---

> > > > > > ### Author Response · Authors · 2024-11-26
> > > > > >
> > > > > > Thank you for your response. We want to make sure we've addressed everything to your satisfaction. Could you please help us understand the specific concern you feel is still unresolved? We're happy to provide more information or discuss this in more detail.

---

### Official Review · Reviewer_LNAn · 2024-11-06

**Soundness:** 2
**Presentation:** 3
**Contribution:** 2
**Rating:** 5
**Confidence:** 2

**Summary:**

This paper presents a machine unlearning framework using parameter-efficient fine-tuning. The approach integrates LoRA-based adapters into the layers of pre-trained networks, which can be trained sequentially or using sequence selection algorithm like BMS. When a data deletion request is made, the associated LoRA parameters tied to the associated data shards are zeroed out, allowing deletions without full retraining. This technique hence enhances deletion efficiency while retaining overall model performance, even after handling multiple unlearning requests.

**Strengths:**

The paper is well-written and easy to follow. The paper is well-written and easy to follow, with a clear explanation of the LoRA-based adapters’ role in isolating model layers for efficient data deletion Validation is performed on a diverse set of tasks and datasets, which strengthens the generalizability performance. The technique effectively handles deletion requests under high budgets, with minimal loss in generalizability.

**Weaknesses:**

W1.	There are some concerns with the proposed top-to-bottom training strategy. First, since different layers are trained with varying data shards, this could cause performance degradation due to inconsistent data exposure across layers. The use of a pretrained base model might mask this potential degradation, as the presented performance relies on the ensemble of pretrained layers with LoRA adapters. It would be beneficial to see results when the model is trained from scratch to evaluate this strategy fully. Additionally, why does a bottom-to-top or other ordering not work well? Some theoretical justification for the chosen order would be helpful.

W2.	It is unclear whether PEFT is essential to this approach or primarily used for computational savings. Could similar outcomes be achieved with additional classification heads or adapters? A discussion on whether PEFT is integral to the technique would be great.

W3.	The novelty of the proposed technique is limited considering the use of off-the-shelf techniques like BMS and LORA to reduce the complexity. A comparison with several recent works like E2URec, Bad-T, NegGrad, NegKL, and RecEraser is missing.

W4.	For a random deletion request unless it is for the slice associated with the final layer, the initial LORA parameters are using not participating for majority of the requests. This raises the question of whether the pretrained model’s representation power makes LoRA adapters in initial layers unnecessary. Any clarifications on this?

W5.	I am a bit confused for the implementation of the baseline SISA. Why the SISA models do not work in your implementation after 40 requests? (In the original paper, the performance does not even visibly drop for tens of deletion requests). It is also mentioned following Lemma 1 that considering real-world scenario, you do not allow retraining of model as originally proposed in SISA. It this is how it is implemented for comparison in the experimental section too?

My score is based on the weaknesses mentioned above. Give clarifications, I am willing to raise my score.

**Questions:**

1.	With higher deletion budgets, the deletion rate does not scale linearly. Would incorporating a retraining strategy improve this? Retraining is mentioned briefly, but further details on when it’s necessary and whether this decision is empirical or rule-based would be helpful.
2.	How does increasing the number of shards and slices impact model accuracy and storage requirements? An ablation study on this would add valuable insight.
3.	A comparison of S3T’s efficiency in terms of time or FLOPs against the baseline would clarify its computational advantages.
4.	In Figure 9, does the SISA baseline include retraining of models, or does it simply use the latest checkpoint? This clarification is necessary for a fair comparison.
5.	How does S3T’s storage requirement compare to SISA’s, particularly with frequent deletion requests?

---

> ### Author Response · Authors · 2024-11-18
>
> We would like to thank the reviewer for their detailed comments and suggestions.
>
> > *Question about top-to-bottom training strategy, the reasoning behind it, and use of pre-trained models.*
>
> The top-to-bottom training strategy in $S^3$T does not use completely different data slices while training each model layer. We take a cumulative approach during training. For example, we train the final layer using slice $S_1$, the next layer is trained data from slices $S_1+S_2$, and this continues. We found that this approach leads to better convergence compared to training different layers separately on just slices $S_1$, $S_2$, and so on.
>
> You are correct that it may not be possible to apply this approach for training a model from scratch. In our work, we assume that the user trains a pre-trained model on private data, as this is the most commonly used setting in modern times. We state this assumption in the problem setting, Lines 144-145.
>
> We did try training models in a bottom-up fashion but it doesn’t converge. It seems that training only the bottom layers while completely freezing all the top layers is not effective for fine-tuning. This is intuitive because several past works [1, 2] have shown that the final layers are affected most during fine-tuning.
>
> [1] What Would Elsa Do? Freezing Layers During Transformer Fine-Tuning. Lee et al. 2019
> [2] How fine can fine-tuning be? Learning efficient language models. Radya-Dixit et al. 2020
>
>
> > *Question about whether PEFT is integral to the technique*.
>
> The choice of the PEFT method is not integral to our fine-tuning technique. This layer-wise training can be performed while using other PEFT (like adapters) or even full-finetuning of the layers. We use LoRA in our work as it is the most popular PEFT technique.
>
>
> We present the results of layer-wise full fine-tuning of ViT-base on various datasets in the table below. We observe that layer-wise finetuning progressively improves performance similar to PEFT methods. It is also able to achieve a similar performance to the overall full-finetuning of the model.
>
>
> Dataset|Stage 1|Stage 2|Stage 3|Stage 4|Stage 5|Stage 6|Full Fine-tuning
> --|--|--|--|--|--|--|--|
> CIFAR-10|97.63|98.10|98.45|98.62|98.75|98.76|98.88|
> CIFAR-100|87.32|89.26|90.35|91.11|91.57|91.87|92.2|
>
> > *Novelty of S3T and comparison with approximate unlearning baselines.*
>
> We would like to clarify that BMS is not an off-the-shelf method. We propose BMS to select sequences with a given deletion prior. BMS uses bipartite matching to select the next element in the constructed sequences. Similarly, layer-wise training and switching off of LoRA layers haven’t been used in the context of exact unlearning.
>
> The mentioned approaches (E2URec, Bad-T, NegGrad, NegKL, RecEraser) are approximate unlearning techniques while our method $S^3$T is an exact unlearning method. These approaches are not directly comparable to our exact unlearning method, $S^3$T.
>
> Approximate unlearning techniques post-process model parameters to ensure unlearning. Since this is not exact, the unlearnt instances can still have a non-zero influence on the model, which is evaluated on benchmarks. In contrast, exact unlearning can guarantee perfect unlearning because it eliminates or retrains all model components that use the unlearnt instance.
>
> Even though the unlearning is perfect, exact unlearning is more expensive than approximate techniques. Therefore, most of the evaluation is focused on the efficiency of unlearning compared to full retraining and other exact unlearning baselines. We provide extensive experiments to evaluate the efficiency and overall model performance in Section 4 and Appendix C.3.
>
> > *Utility of initial LORA layers*
>
> You are correct that if the deletion requests are not affecting the final layers the initial LORA layers are not impacted and therefore are not switched off. However, these layers are still being used for performing inference and are pivotal for the overall model performance.
>
> Please let us know if you have any further details or additional questions.
>
> > *Details of SISA implementation*
>
> We consider the setting where SISA falls back to the best available checkpoint instead of retraining after each deletion request. We use this setting for all baselines and experiments. This setup is realistic since taking down a production system for retraining can affect customers and disrupt business operations. This also helps us measure the deletion rate – the expected number of deletion requests a system can handle without requiring to be retrained.
>
> SISA stops working after 40 deletion requests because it does not have any available checkpoints. This is different from the implementation in [3], where SISA components are retrained after every deletion request. We observe that $S^3$T can handle significantly more deletion requests while incurring a lesser performance impact.
>
> [3] Machine Unlearning, Bourtoule et al. 2020

---

> ### Author Response · Authors · 2024-11-18
>
> > *Would incorporating a retraining strategy improve the deletion rate? Question about the necessity of retraining.*
>
> Yes, your observation is correct that the deletion rate does not scale linearly.
>
> In fact, increasing the budget $B$ beyond the number of slices $L$ doesn’t improve the deletion rate at all. We prove this in Lemma 1 (where the deletion rate is a function of $\min(B, L)$) and show this linear growth empirically in Figure 6 (right). Even though budgets $B>L$ don’t improve the deletion rate they help improve the performance of the available models (shown in Lemma 2).
>
> Retraining after a certain point will help improve the performance. However, frequent retraining should be avoided, as multiple deletion requests affecting the same data slice could lead to redundant retraining of the same layer. The user should set a performance threshold based on their application and retrain the components only when the overall performance drops below it.
>
> > *How does increasing the number of shards and slices impact model accuracy and storage requirements? An ablation study on this would add valuable insight.*
>
> We perform an experiment to investigate the impact of shard count. We train ViT-base using different shard counts on the CIFAR100 dataset. In the table below, we observe a slight decline in performance as the number of shards increases. This is expected as each model is trained on a smaller number of instances. This observation is similar to the observation made by the authors in [1] (see Figure 6 in [1]).
>
> Number of Shards|Performance|
> --|--|
> 1|90.67|
> 2|89.54|
> 3|88.62|
> 4|87.72|
> 5|87.11|
>
> We conducted experiments to investigate the impact of the number of slices (and the number of PEFT layers allocated to each slice) on model performance, as detailed in Figure 13 (Appendix C.2). We observe that using either too many or too few slices negatively impacts performance. Ideally, we should select a balanced setting to optimize performance using this experiment. Additionally, the number of slices doesn’t have any impact on the storage as all the updates are stored in the same model.
>
> [3] Machine Unlearning, Bourtoule et al., 2020
>
> > *A comparison of S3T’s efficiency in terms of time or FLOPs against the baseline would clarify its computational advantages.*
>
> We have reported these results in the main paper. Please find the unlearning time comparison in Figure 9 (Section 4).
>
> > *In Figure 9, does the SISA baseline include retraining of models, or does it simply use the latest checkpoint? This clarification is necessary for a fair comparison.*
>
> Figure 9 does include the retraining time of models. The small steps in the plot for $S^3$T and SISA indicate the retraining stages where the time required is more. The flat portions of the curve indicate the checkpoint-swapping phase.
>
> We report the exact numbers for the plot below:
>
> Average re-training time for $S^3$T: 4.32 hours
>
> Average re-training time for SISA: 4.85 hours
>
> Checkpoint swapping time (same for both): 1.62 seconds
>
> > *How does S3T’s storage requirement compare to SISA’s, particularly with frequent deletion requests?*
>
> The storage requirements of these systems are independent of the deletion requests. They are a function of the number of shards ($m$), slice count ($L$), and budget ($B$).
>
> SISA: $mL$ (for each of the $m$ shards, one checkpoint after training each of $L$ slices)
>
> $S^3$T: $mB$ (for each of the $m$ shards one checkpoint the allocated budget $B$)
>
> Note that $S^3$T doesn’t need to store a checkpoint after training each slice as they are stored in the PEFT layers of the same model.

---

> > ### Author Response · Authors · 2024-11-20
> >
> > Thanks for taking the time to review our responses. Please let us know if you have any questions or would like any further clarifications. If your questions have been answered, we kindly ask that you consider adjusting your score. Thank you!

---

> > > ### Author Response · Authors · 2024-11-24
> > >
> > > Thank you for reviewing our responses again. Since the discussion period ends soon, we wanted to double-check whether your questions have been addressed. If you need any further clarification, please let us know.

---

> > > > ### Author Response · Authors · 2024-11-25
> > > >
> > > > Since the rebuttal period is drawing to a close, we wanted to check in again and see whether our rebuttal has addressed your comments. Please let us know if we can answer any other questions you have.

---

> > > > > ### Author Response · Authors · 2024-12-01
> > > > >
> > > > > Dear Reviewer,
> > > > >
> > > > > We are reaching out one final time before the discussion period concludes tomorrow. We've put significant effort into addressing your questions and would appreciate your feedback. If your concerns have been addressed, please consider updating the score accordingly. We are more than happy to discuss any remaining concerns that you may have. Thank you!

---

> > > > > > ### Comment · Reviewer_LNAn · 2024-12-02
> > > > > >
> > > > > > Thank you for your clarifications and detailed responses. While your explanations addressed some of my queries, I still find the significance of the work to be marginal. Nevertheless, I acknowledge the effort and rigor in your work, and I have raised my score by 2 points accordingly.

---

### Author Response · Authors · 2024-11-18
**General Response**

We thank the reviewers for taking the time to go through our paper and provide detailed feedback/suggestions. We have conducted additional experiments and revised the paper to address any concerns. Here is the summary of all the changes:


* We evaluate that our proposed layer-wise training approach is effective when performing full-finetuning of the layers.
* We conducted an experiment to evaluate the impact of the number of shards on the model performance.
* We added an explanation of why exact unlearning techniques provide stronger privacy guarantees in Section 2.
* We have added a new experiment to evaluate the deletion rate of $S^3$T when the deletion requests are sampled from a non-uniform prior (Appendix C.3).
* We have updated Figure 1 and the corresponding caption in the draft.
* We have incorporated the conclusions of RQ2 and RQ3 in Section 4.2 and Section 4.3 respectively.
* We have provided several ablations of the baseline in Appendix C.3 (Baseline Ablations).
* We conducted experiments to evaluate the impact of using an ensemble of models trained on different slice sequences in Appendix C.3 (Ensemble of Slice Sequences).
* We have included a limitation and future directions section in Appendix D of the paper.

---

### Meta-Review · Area_Chair_7dWz · 2024-12-19

**Metareview:**

This work proposes a method for exact unlearning that a relies on learning (and combining) multiple models on disjoint data shards. Learning of the models leverages parameter efficient fine-tuning to enable parameter isolation. The methodology is clearly described and positioned with respect to the state of the art. The authors produced compelling experimental evidence to validate the approach, suggesting strong performances. They also discussed the computational complexity of the method. Most reviewers voted for acceptance and all were satisfied with the clarifications provided by the authors during rebuttal.

**Additional Comments On Reviewer Discussion:**

Reviewers acknowledged the response provided by the authors. Several adjusted their scores based the detailed clarifications, including producing additional empirical evidence. Post rebuttal, one reviewer indicated that they found the contributions incremental and voted for a weak reject as a result. However, they did not further elaborate on this point.

---

### Decision · Program_Chairs · 2025-01-22

Accept (Poster)